# On the Parameterization of Second-Order Optimization Effective Towards the Infinite Width

**Satoki Ishikawa**
Department of Computer Science
Tokyo Institute of Technology, Japan
`riverstone@rio.gsic.titech.ac.jp`

**Ryo Karakida**
Artificial Intelligence Research Center
AIST, Japan
`karakida.ryo@aist.go.jp`

## Abstract

Second-order optimization has been developed to accelerate the training of deep neural networks and it is being applied to increasingly larger-scale models. In this study, towards training on further larger scales, we identify a specific parameterization for second-order optimization that promotes feature learning in a stable manner even if the network width increases significantly. Inspired by a maximal update parameterization, we consider a one-step update of the gradient and reveal the appropriate scales of hyperparameters including random initialization, learning rates, and damping terms. Our approach covers two major second-order optimization algorithms, K-FAC and Shampoo, and we demonstrate that our parameterization achieves higher generalization performance in feature learning. In particular, it enables us to transfer the hyperparameters across models with different widths.

## 1 Introduction

Second-order optimization has been attracting considerable interest in improving the training efficiency of neural networks (Amari, 1998; Pascanu & Bengio, 2014). It accelerates the convergence of gradient dynamics (Martens & Grosse, 2015; Gupta et al., 2018; Li, 2017) and can optimize neural networks that are especially hard to train due to highly distorted parameter landscape (Martens et al., 2018; Zhang et al., 2022a; Beyer et al., 2022). Because of these successes in improving training efficiency, it is increasingly being applied to even larger-scale models year by year (Osawa et al., 2023b; Pauloski et al., 2021; Shi et al., 2021; Zhang et al., 2023; 2022b; Anil et al., 2021).

In general, gradient methods depend on some hyper-parameters (HPs), including a learning rate, and we need careful HP tuning to achieve better performance. The most straightforward approach is to train the model multiple times and search for better HP, but for large-scale models, the computational cost is high even for a single training, making it very challenging to find it. In particular, second-order optimization methods possess not only a learning rate but also a damping term which requires careful tuning (Pascanu & Bengio, 2014; Martens, 2020).

How can we obtain quantitative insight into appropriate settings of HP that hold on large scales? Usually, an appropriate scale of HPs depends on the model size and we need to scale them up or down depending on the network width or depth (Park et al., 2019; Iyer et al., 2023; Dinan et al., 2023). One approach to obtaining a robust scale of HPs that universally works in large models is to consider a large limit of the model size (Schoenholz et al., 2017; Lee et al., 2018; Xiao et al., 2018; Luo et al., 2021). In particular, to obtain a preferable HP that works in first-order gradient descent, Yang & Hu (2021) proposed Maximum Update Parameterization (MUP; $\mu$P). They analyzed an infinite width limit of neural networks and successfully identified HPs that prevent the parameter update from vanishing or exploding and ensured stable progress of training, which is independent of the width. The $\mu$P also enables us to avoid the lazy regime, where the model parameters remain sufficiently close to the initialization and allow feature learning at the maximum scale of the update in all layers. Moreover, Yang et al. (2021) have demonstrated that since the $\mu$P works effectively towards the infinite width, we can re-use the HP including a learning rate obtained in a smaller model to the large-scale one. To date, however, these studies have been limited to first-order and entry-wise optimization (Yang & Littwin, 2023).

In this work, we consider an infinite width limit and propose a $\mu$P for second-order optimization including two major methods; K-FAC (Martens & Grosse, 2015) and Shampoo (Gupta et al., 2018). Our main contributions can be summarized as follows:

- We consider a one-step update of the second-order optimization methods in the infinite-width limit and reveal the HPs (i.e., scales of random initialization, learning rates, damping terms) that allow the $\mu$P (in Section 4.1). We especially clarify that the stable feature learning requires specific scales of learning rates; K-FAC works for constant learning rates whereas Shampoo requires scaling depending on the width. Regarding the damping terms, we find that a classical heuristic scaling of Shampoo satisfies the $\mu$P condition while that of K-FAC requires re-scaling (in Section 4.2).

- In practice, the last layer's weight is sometimes initialized not by random initialization but by zero. By carefully considering the one-step update, we find that while the zero initialization allows feature learning in the usual first-order gradient, it can cause an approach to the network parameter corresponding to the neural network Gaussian process (NNGP) in the case of K-FAC (in Section 4.3). This can be regarded as a novel implicit bias specific to K-FAC that appears in the infinite-width limit.

- We empirically verify the effectiveness of our proposed parameterization in the training of various neural networks. In particular, it enables us to transfer optimal learning rates and damping terms from narrow models to wider ones (in Section 5.2, 5.3).

Thus, this work provides quantitative insights that will serve as a foundation for scaling second-order optimization towards the learning of even larger models in the future.

## 2 RELATED WORK

**Feature learning:** In general, it is highly non-trivial that the large limit of the model allows stable learning under the widely-used standard parameterization (SP). In the infinite width limit, we may have unstable dynamics (i.e., vanishing/exploding gradient depending on the width) or, more non-trivially, the lazy regime (a.k.a. neural tangent kernel regime) (Chizat et al., 2019; Jacot et al., 2018). While the learning dynamics in the lazy regime progress in a stable manner, the parameters remain sufficiently close to the initialization, and the network is essentially approximated by a linear model. This means that no feature learning appears, thus there is growing interest in under what conditions feature learning progresses outside of the lazy regime (Woodworth et al., 2020; Geiger et al., 2021; Luo et al., 2021; Bordelon & Pehlevan, 2022). Based on an order evaluation of parameter updates in the middle of training, Yang & Hu (2021) proposed $\mu$P for stochastic gradient descent (SGD) that realizes feature learning in the infinite-width limit. Some experimental studies demonstrated the utility of $\mu$P Yang et al. (2021); Vyas et al. (2023). The theory is also generalized to entry-wise adaptive gradient methods including Adam (Littwin & Yang, 2023). The second-order optimization does not belong to the class analyzed in these existing studies, and thus the current work is the first to challenge this problem. Note that the second-order optimization in the lazy regime has been investigated by some work (Zhang et al., 2019a; Cai et al., 2019; Karakida & Osawa, 2020).

**Second-order optimization:** The preconditioned gradient can speed up the convergence of training dynamics. Natural gradient descent (NGD) (Amari, 1998) is a classical example of second-order optimization which preconditions the gradient by the inverse of the Fisher information matrix. However, since the computation of the inverse is computationally demanding, some approximation is required. K-FAC (Martens & Grosse, 2015; Grosse & Martens, 2016) is such an approximation for deep neural networks and it has been frequently employed for training large-scale models where acceleration is particularly important (Osawa et al., 2023b; Pauloski et al., 2020; 2021; Shi et al., 2021; Zhang et al., 2023; 2022b). Shampoo is another commonly used second-order optimization method (Gupta et al., 2018) and achieves fast convergence (Anil et al., 2021). Note that second-order optimization methods contain a damping term. Careful selection of such HPs is known to be important for the success of training (Martens, 2020; Zhang et al., 2019b; Gao et al., 2020).

## 3 PRELIMINARIES

This section explains the second-order optimization in $L$-layered fully-connected neural networks:

$$\boldsymbol{u}_l = \boldsymbol{W}_l \boldsymbol{h}_{l-1} + \boldsymbol{b}_l, \quad \boldsymbol{h}_l = \phi(\boldsymbol{u}_l) \quad (l = 1, ..., L), \tag{1}$$

where we define weight matrices $\boldsymbol{W}_l \in \mathbb{R}^{M_l \times M_{l-1}}$, bias terms $\boldsymbol{b}_l$, and activations $\boldsymbol{h}_l \in \mathbb{R}^{M_l}$. We set the width of the hidden layer to $M_l = M$ ($l = 1, ..., L - 1$) for clarity, but this does not lose the generality of the following analysis. $\phi(\cdot)$ is a differentiable and polynomially-bounded activation function and theoretical works in $\mu$P usually assume either Tanh function or $\sigma$-GELU function if necessary (Yang & Hu, 2021). Let $(\boldsymbol{x}_i, \boldsymbol{y}_i)$ be a pair of input and target training sample. For simplicity, we consider the mean squared loss function for a one-dimensional target: $\mathcal{L}(\boldsymbol{\theta}) = \frac{1}{n} \sum_{i=1}^{n} \|y_i - f_{\boldsymbol{\theta}}(\boldsymbol{x}_i)\|^2 \in \mathbb{R}$, where $\boldsymbol{\theta}$ denotes a vector of all parameters and $f_{\boldsymbol{\theta}} = u_L$ is the output of the deep neural network. It is straightforward to generalize the following results to the cases of multi-classes and the cross-entropy loss as mentioned in Section A.5.

### 3.1 OVERVIEW OF SECOND-ORDER OPTIMIZATION

Second-order optimization is an algorithm that updates parameters by a preconditioned gradient: $\boldsymbol{\theta}_{t+1} = \boldsymbol{\theta}_t - \eta (\boldsymbol{C}(\boldsymbol{\theta}_t) + \rho \boldsymbol{I})^{-1} \nabla_{\boldsymbol{\theta}_t} \mathcal{L}(\boldsymbol{\theta}_t)$, where $\eta$ is a learning rate, $\boldsymbol{C}(\boldsymbol{\theta})$ is the curvature matrix and $\rho$ is the damping term. Usually, this inverse is computationally demanding and hard to use. Therefore, the following seminal works have introduced smaller preconditioners for each layer and updated rules in a matrix form.

**K-FAC:** Natural gradient descent (NGD) is the case where $\boldsymbol{C}$ is given by the Fisher information matrix. K-FAC approximates this $\boldsymbol{C}$ by the Kronecker product of two matrices (Martens & Grosse, 2015; Grosse & Martens, 2016). Its update rule in the matrix form is given by

$$\boldsymbol{W}_{l,t+1} = \boldsymbol{W}_{l,t} - \eta (\boldsymbol{B}_l + \rho_{B_l} \boldsymbol{I})^{-e_B} \nabla_{\boldsymbol{W}_l} \mathcal{L}(\boldsymbol{\theta}_t)(\boldsymbol{A}_{l-1} + \rho_{A_{l-1}} \boldsymbol{I})^{-e_A}, \tag{2}$$

where $e_A = e_B = 1$. The preconditioning matrices are given by $\boldsymbol{B}_l = \mathbb{E}[\boldsymbol{\delta}_l \boldsymbol{\delta}_l^\top]$ and $\boldsymbol{A}_l = \mathbb{E}[\boldsymbol{h}_l \boldsymbol{h}_l^\top]$ where $\boldsymbol{\delta}_l = \nabla_{\boldsymbol{u}_l} f_{\boldsymbol{\theta}}$ and $\mathbb{E}[\cdot]$ is an average over training samples. General exponents $e_A$ and $e_B$ are introduced here because some work only considers to use a part of preconditioners like $(e_A, e_B) = (1, 0)$ (Benzing, 2022; Amid et al., 2022). The size of damping terms is usually determined in a heuristic manner as mentioned in Section 6.3.

**Shampoo:** Gupta et al. (2018) proposed the following update rule as a second-order optimization;

$$\boldsymbol{W}_{l,t+1} = \boldsymbol{W}_{l,t} - \eta (\boldsymbol{L}_l + \rho_{L_l} \boldsymbol{I})^{-e/2} \nabla_{\boldsymbol{W}_l} \mathcal{L}(\boldsymbol{\theta}_t)(\boldsymbol{R}_{l-1} + \rho_{R_{l-1}} \boldsymbol{I})^{-e/2}, \tag{3}$$

where $\boldsymbol{L}_l = \mathbb{E}[\boldsymbol{\delta}_l \boldsymbol{h}_{l-1}^\top \boldsymbol{h}_{l-1} \boldsymbol{\delta}_l^\top]$ and $\boldsymbol{R}_l = \mathbb{E}[\boldsymbol{h}_{l-1} \boldsymbol{\delta}_l^\top \boldsymbol{\delta}_l \boldsymbol{h}_{l-1}^\top]$ with $\boldsymbol{\delta}_l = \nabla_{\boldsymbol{u}_l} \mathcal{L}$. In Shampoo, $e = 1/2$ is applied. If we neglect the non-diagonal entries of the preconditioners, this is similar to Adam and AdaGrad.

### 3.2 ABC-PARAMETERIZATION

ABC-parameterization scales parameters by the width as follows (Yang & Hu, 2021):

$$\boldsymbol{W}_l = \boldsymbol{w}_l / M^{a_l}, \quad \boldsymbol{w}_l \sim \mathcal{N}(0, \sigma'^2 / M^{2b_l}), \quad \eta_l = \eta'_l / M^{c_l}, \tag{4}$$

The $\mu$P is an abc-parameterization that induces feature learning in every layer in a model with infinite widths. In short, the previous work characterizes *feature learning* by

$$\Delta \boldsymbol{h}_l := \boldsymbol{h}_{l,t} - \boldsymbol{h}_{l,0} = \Theta(1)^1, \tag{5}$$

where $\Theta(\cdot)$ denotes the Big Theta notation for the order evaluation with respect to the width. Note that for the lazy regime, we have $\Delta \boldsymbol{h}_l = o(1)$ in hidden layers. The previous work found that the feature learning (5) appears for some specific conditions. In particular, the following condition of $W_l$ *updated maximally* plays a fundamental role:

$$\Delta \boldsymbol{W}_l \boldsymbol{h}_{l-1} = \Theta(1). \tag{6}$$

---

[1]Precisely speaking, this is feature learning (Definition H.9) especially satisfying a stability condition (Theorem H.6) in Yang & Hu (2021). Note that as in the previous work, the last equality represents *the coordinate size* where $v = \Theta(M^a)$ means $\sqrt{\|v\|^2 / M} = \Theta(M^a)$ for $v \in \mathbb{R}^M$

Table 1: $\mu$P for K-FAC and Shampoo.

|  | Input weights & all biases | Output weights | Hidden weights |
|---|---|---|---|
| SP | $b = 0, c = 0$ | $b = 1/2, c = 0$ | $b = 1/2, c = 0$ |
| SGD ($e = 0$) | $b = 0, c = -1$ | $b = 1, c = 1$ | $b = 1/2, c = 0$ |
| Shampoo ($e = \frac{1}{2}$) | $b = 0, c = -1/2$ | $b = 1, c = 1/2$ | $b = 1/2, c = 0$ |
| K-FAC ($e_{A,B} = 1$) | $b = 0, c = 0$ | $b = 1, c = 0$ | $b = 1/2, c = 0$ |

Yang & Hu (2021) analyzed feature learning in the one-step gradient under this condition. Note that they also obtain a mathematical expression of forward and backward propagated signals for the first-order gradient at general time steps by the Tensor Program. For the derivation of the $\mu$P, we focus only on the first one-step gradient as is explained in Section A.2 of Appendices.

In the following sections, we will set $a_l = 0$ for all layers in the same way as in the first-order case (Yang et al., 2021). The previous work has demonstrated that the $\mu$P of the first-order gradient is scale-invariant to the constant shift $(a, b, c) \leftarrow (a, b, c) + (k, -k, -2k)$. As we will show later, the $\mu$P of second-order optimization also has this indeterminacy and we can eliminate it by $a_l = 0$.

## 4 PREFERABLE SCALING OF HPS IN SECOND-ORDER OPTIMIZATION

### 4.1 $\mu$P FOR SECOND-ORDER OPTIMIZATION

In this section, we derive the $\mu$P by considering the first one-step update of second-order optimization. We suppose that the damping term $\rho_X$ ($X = \{A, B, L, R\}$) satisfies

$$\rho_X = \rho'_X / M^{d_X}, \tag{7}$$

with a positive constant $\rho'_X$. For simplicity, suppose a one-step update from the initialization where the gradient and all preconditioners are computed on the same samples (Assumption A.5). We also suppose common assumptions used in $\mu$P for the first-order gradient (Assumptions A.3,A.4). When the eigenvalues of the preconditioning matrices have an equal width-dependent scale to the damping term (e.g., $\rho_A = \Theta(\|A\|_2)$), we say that the second-order optimization is *valid* (Definition A.6). Then, we obtain the following.

**Proposition 4.1 ($\mu$P of second-order parameterization).** *Consider the first one-step update of K-FAC and Shampoo in the infinite width limit. The second-order optimization becomes valid for*

$$d_{A_l} = \begin{cases} -1 & 1 < l \leq L \\ 0 & l = 1 \end{cases}, \quad d_{B_l} = \begin{cases} 0 & l = L \\ 1 & 1 \leq l < L \end{cases}, \quad d_{L_l}, d_{R_l} = \begin{cases} 1 & l = 1 \\ 0 & 1 < l < L \\ -1 & l = L \end{cases}. \tag{8}$$

*It admits the $\mu$P for feature learning at*

$$b_l = \begin{cases} 0 & l = 1 \\ 1/2 & 1 < l < L \\ 1 & l = L \end{cases}, \quad c_l = \begin{cases} e_B - 1 & l = 1 \\ e_B - e_A & 1 < l < L \\ 1 - e_A & l = L \end{cases}, \tag{9}$$

*where we set $a_l = 0$ and setting $e_A = e_B = e$ corresponds to Shampoo.*

*Rough sketch of derivation.* The detailed and comprehensive derivation is presented in Section A. Here, let us briefly explain the derivation of $\mu$P for K-FAC. The $\mu$P has two conditions to be satisfied (A.1,A.2). For an infinitesimal one-step update, these conditions have explicit and tractable expressions as described in Section A.2. They are applicable to both first-order and second-order optimization methods. To check the conditions, we use the push-through identity:

$$\Delta W_l h_{l-1} = 1/M^{2a+c}(B + \rho_B I)^{-e_B} \delta \text{diag}(\chi) h^\top (A + \rho_A I)^{-e_A} h$$

$$= 1/M^{2a+c} \delta (\delta^\top \delta + \rho_B I)^{-e_B} \text{diag}(\chi)(h^\top h + \rho_A I)^{-e_A} h^\top h \tag{10}$$

where we omitted the layer index and $\chi$ means an error vector. In the second line, the size of the inverse matrix is independent of the width and this expression enables us to carry out the order evaluation of $\Delta W_l h_{l-1}$. Then, $\mu$P's conditions become

$$2a_1 + c_1 + e_B - (2e_B - 1)(a_L + b_L) = 0, \tag{11}$$

$$2a_l + c_l + e_A + e_B - 1 - (2e_B - 1)(a_L + b_L) = 0, \tag{12}$$

$$2a_L + c_L + e_A - 1 = 0, \quad a_L + b_L - 1 = 0. \tag{13}$$

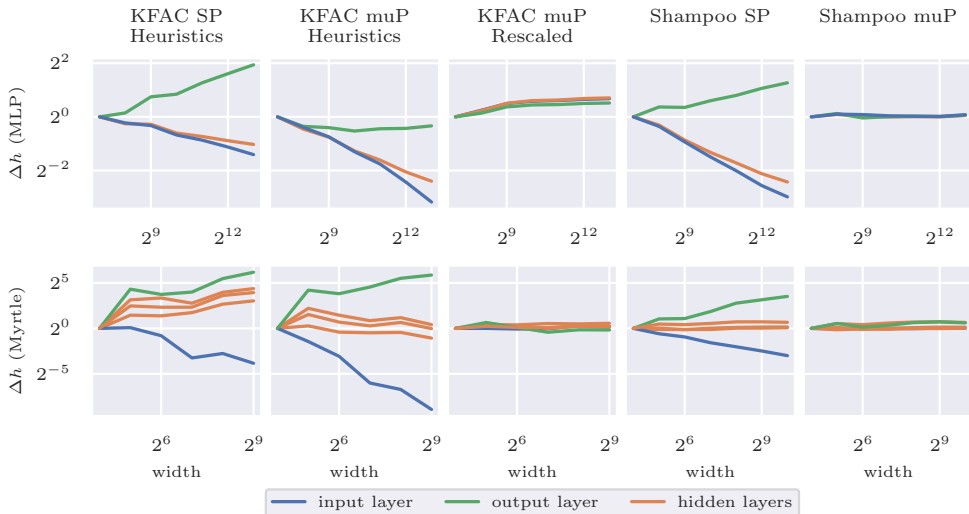

**Figure 1: $\mu$P achieves feature learning across the width.** In SP (Pytorch's Default), $\Delta h_l$ in each layer exhibits dependence on the width. For K-FAC, the default setting of the damping (heuristics) does not satisfy the condition of $\mu$P and we need to utilize the rescaled one as is explained in Section 4.2. We train 3-layer MLP with CIFAR10 in the first line and Myrtle-5 with CIFAR10 in the second line. This result does not depend on exponential moving averages or activation (Appendix.D).

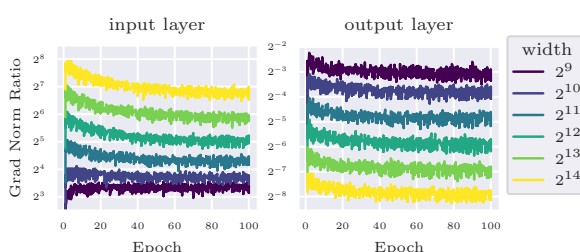

**Figure 2: The obtained parameterization is consistent throughout the training.** The order of the curvature matrix in K-FAC does not change with time. The input layer is proportional to $M$ whereas the output layer is proportional to $1/M$, which is a natural order in terms of the $\mu$P of the SGD. We trained a 3-layer MLP on FashionMNIST dataset.

Fixing a constant shift by setting $a_l = 0$, we obtain the result. □

Here, let us explain some rather technical details of the result. First, note that we derived the $\mu$P from the one-step update. This is the same as in the original work on $\mu$P where the one-step update determines the $\mu$P for the whole (inductive) $t$-step updates. In fact, we empirically confirm the effectiveness of $\mu$P for $t > 1$. Second, we focus on the case where the preconditioning matrices become valid. Even if $\rho$ takes a much larger value than these matrices, the gradient may realize the feature learning under an appropriate scaling because it reduces to the first-order gradient. However, such unusual switching between the first and second-order optimization is outside the scope of the current work. Thus, we refer to the setting of this proposition as $\mu$P of second-order parameterization.

Table 1 summarizes the $\mu$P for K-FAC and Shampoo. $b_L = 1$ is a common characteristic of $\mu$P for all methods whereas $c_1$ and $c_L$ depend on $e$. One interesting point is that just by setting $b_L = 1$, K-FAC with $\mu$P works for $c_l = 0$. That is, *K-FAC does not require scaling of the learning rate* in contrast to the first-order gradient (SGD), which requires $c_l$ depending on the width. Moreover, Eq.(9) indicates that this is unique to the K-FAC ($e = 1$). In other words, K-FAC's preconditioning effectively achieves the $\mu$P's scaling of learning rates of the first-order gradient. Note that we can also extend $\mu$P to the Gauss-Newton method without using the K-FAC approximation (Appendix A.4.3). As a side note, we also summarize the parameterization for the lazy regime in Section I.

Figure 1 empirically confirms that $\mu$P realizes the feature learning (5) in the training of MLP and Myrtle-5. The order of the features was kept during the training, as is shown in Figure 2. Figure 1 also justifies the $\mu$P in CNN. In the CNN model, width represents the number of channels (Xiao et al., 2018; Yang et al., 2021). Although our $\mu$P is derived on the MLP and K-FAC for CNN includes

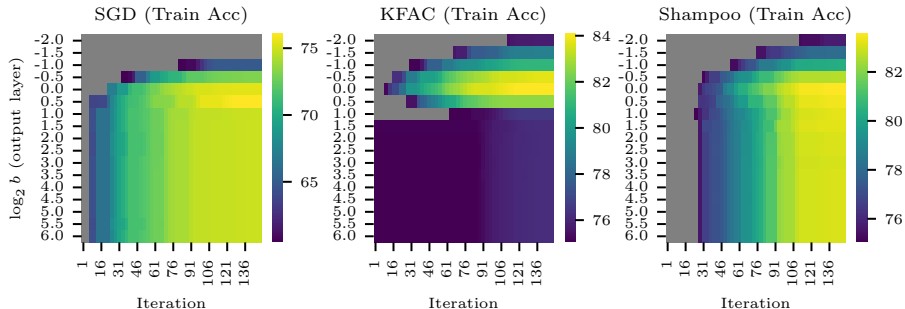

**Figure 3: K-FAC converges to the NNGP solution when the variance of the last layer is close to zero.** When $b_L$ in the last layer is increased (other parameters are fixed to $\mu$P), K-FAC can converge to the NNGP solution in one step. Therefore, when $b_L$ is increased, it converges to a kernel solution, which limits $b$ in which feature learning can occur.

an additional approximation in addition to K-FAC for MLP, we empirically observe that the same parameterization as MLP is also true for CNN.

## 4.2 JUSTIFICATION OF DAMPING HEURISTICS

Implementations in practice often adjust the damping term of K-FAC using the following heuristics:

$$\rho_{A_{l-1}}(= 1/\rho_{B_l}) := \sqrt{\frac{\operatorname{tr}(\boldsymbol{A}_{l-1})}{M} \frac{M}{\operatorname{tr}(\boldsymbol{B}_l)}} \rho' = O(\sqrt{M \cdot M^{2(a_L + b_L) - 1}}), \tag{14}$$

that is $O(M)$ for the $\{a, b, c\}$ given in Proposition 4.1. This heuristics is consistent with the valid damping scales (8) in hidden layers but *not in the input and output layers*. This causes $\Delta\boldsymbol{h}_l$ to decay when using damping heuristics even if $a, b, c$ are set to $\mu$P settings. See Section A.6 for more details. To overcome this problem, we propose to use the following *rescaled damping* satisfying the valid damping scale in $\mu$P:

$$\rho_A^{Re} := \rho' \operatorname{tr}(\boldsymbol{h}_{l-1}^\top \boldsymbol{h}_{l-1}), \quad \rho_B^{Re} := \rho' \operatorname{tr}(\boldsymbol{\delta}_l^\top \boldsymbol{\delta}_l). \tag{15}$$

This rescaling is useful because it enables explicitly expressing the dampings of all layers in a unified manner. Furthermore, it provides the heuristic scales of the preconditioners (i.e., trace) as proportional coefficients which are expected to ensure stable learning dynamics. If damping is set consistent with $\mu$P, $\Delta\boldsymbol{h}_l$ neither decays nor grows with respect to width as shown in Figure 1 (3rd column).

In contrast to the K-FAC, we found that a standard heuristics of the damping, i.e., a constant multiple of the largest eigenvalue of $\boldsymbol{L}_l, \boldsymbol{R}_l$, is consistent with the $\mu$P in Proposition 4.1 and requires no modification towards the infinite width. See Section A.6 for more detail.

## 4.3 IMPLICIT BIAS OF K-FAC TOWARDS NNGP AT ZERO INITIALIZATION

Up to here, we supposed the most common setting where the output weight is given by random (Gaussian) initialization. However, some recent implementations utilized zero initialization on the head (Yang et al., 2021; Wightman, 2019; Botev & Martens, 2022). This corresponds to $b_L = \infty$, which also induces feature learning after the second step in SGD. The reason why feature learning occurs even at $b = \infty$ can be explained as follows. When the last layer is initialized with zero, the weights after a 1-step SGD update are

$$\boldsymbol{W}_{l,t=1} = \eta_l \boldsymbol{h}_{l,0} \boldsymbol{y} \ (l = L), \quad \boldsymbol{W}_{l,0} \ (l < L). \tag{16}$$

Since $\boldsymbol{W}_{L,t=1} = \Theta(1/M)$, the weights after a 1-step update can be regarded as weights initialized by $\mu$P. Thus feature learning begins from the second step. This is also true for $b_L > 1$. However, in K-FAC, when the last layer is initialized with zero, a single update results in the weight:

$$\boldsymbol{W}_{l,t=1} = \eta_l (\boldsymbol{h}_{l-1,0} \boldsymbol{h}_{l-1,0}^\top + \rho_A \boldsymbol{I})^{-1} \boldsymbol{h}_{l,0} \boldsymbol{y} \ (l = L), \quad \boldsymbol{W}_{l,0} \ (l < L). \tag{17}$$

Interestingly, this represents an NNGP solution (Lee et al., 2018). When the model can deviate from the NNGP solution, feature learning begins in the training process, similar to SGD. However, when

| Optimizer | $b_L$ | Batch Size | | | | | Table 2: |
|---|---|---|---|---|---|---|---|
| | | 4 | 16 | 64 | 256 | 1024 | |
| SGD | 0.5 | 82.60 | 80.91 | 78.86 | 74.50 | 66.83 | |
| | 1.0 | 83.62 | 83.61 | 82.10 | 77.99 | 73.40 | |
| | 64.0 | **83.98** | **83.82** | **82.60** | **79.53** | **74.63** | |
| K-FAC | 0.5 | 83.18 | **84.17** | 83.75 | 84.07 | 80.25 | |
| | 1.0 | **83.84** | 84.16 | **84.29** | **84.33** | **83.21** | |
| | 64.0 | 81.56 | 82.72 | 82.63 | 79.51 | 75.37 | |

**Table 2: Test accuracy for different batch sizes shows the consistent results** (3-layer CNN on FashionMNIST. The whole dataset size is set to 1024). The case of $b_L = 1$ consistently performs better than $b_L = 0.5$ and $b_L = 64$ in K-FAC but not in SGD. Learning rates are tuned for each batch size.

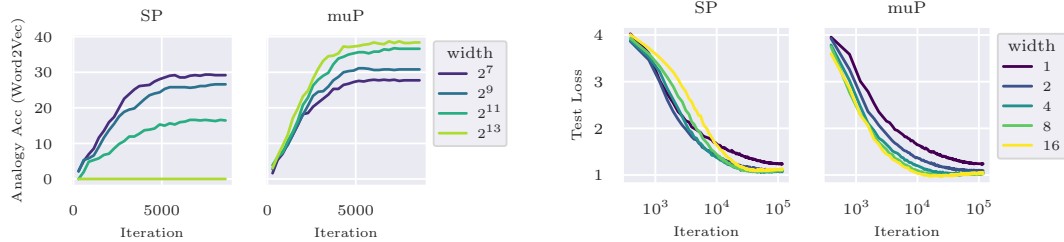

**Figure 4: Wider models learn well under $\mu$P throughout training.** Using $\mu$P, training proceeds equally across widths. In $\mu$P, the loss is lower for wider widths throughout training. (Left) We trained CBOW on WikiText2 by Shampoo with various widths. (Right) We trained ResNet18 on CIFAR100 by K-FAC while increasing the number of channels from 1 to 16.

the batch size is close to the full batch size or the learning rate is set considerably low, moving away from the NNGP solution becomes challenging and does not incur feature learning. Since K-FAC is often used for large batch training, we must be careful about its behavior on $b_L \gg 1$.

Figure 3 confirms this phenomenon. In this experiment, we trained a 3-layer CNN on FashionMNIST with MSE loss. For simplicity, the dataset size is reduced to 1024 and we are training in full batch. After sufficient learning, the training accuracy is highest only for $b_L = 1$. Since the parameters are fixed at the initial kernel solution when $b_L \gg 1$, we can observe that accuracy at $b_L \gg 1$ is lower than $b_L = 1$. However, SGD and Shampoo achieve almost the same accuracy for $b_L = 1$ and $b_L \gg 1$. Table 2 shows that when $b_L \gg 1(b_L = 64)$, its accuracy on K-FAC is consistently lower than $b_L = 1$ while there is no decrease in accuracy by setting $b_L = 64$ in SGD. In addition, $b_L = 1$ ($\mu$P) achieves higher accuracy than $b_L = 0.5$ (SP) across different batch sizes. We empirically observed that the difference in accuracy between $b_L = 1$ and $b_L = 64$ decreases as the batch size decreases. The same behavior can also be observed when using cross-entropy loss (Appendix.E.2).

Note that our purpose is not to fully deny the current usage of zero initialization in relatively finite neural networks. Our finding claims that the current default settings do not necessarily work well with large models, and careful attention is required.

## 5 EXPERIMENTS

### 5.1 $\mu$P IN WIDE NEURAL NETWORKS

$\mu$P can invoke feature learning equally across width. This enables *a wider model performs better* when trained in $\mu$P and given the same HPs (Yang et al., 2021). This section demonstrates that the above statement is also true for second-order optimization.

**Word2Vec:** This is a toy model used in the previous work of $\mu$P (Yang & Hu, 2021) to check the advantage of feature learning. We train CBOW on Wikitext2 dataset with Shampoo and evaluate its embedding vectors by a Word Analogy task[2]. Figure 4 shows that in SP the accuracy decreases as the width is increased. In the infinite width limit of SP, the embedding layer is fixed at initialization, and the accuracy does not increase from the churn rate. However, in $\mu$P, increasing the width does not

---

[2]K-FAC is very sensitive to $b_L$, and we found it difficult to optimize word2vec in K-FAC. This could be explained by Section 4.3. Its details are described in Appendix E.5.

| | VGG19 (C100) | | | | ResNet18 (C100) | | | | ResNet50 (INet) | |
| | K-FAC | | Shampoo | | K-FAC | | Shampoo | | K-FAC | |
| width | SP | $\mu$P | SP | $\mu$P | SP | $\mu$P | SP | $\mu$P | SP | $\mu$P |
|---|---|---|---|---|---|---|---|---|---|---|
| 1 | 64.96 | +0.31 | 63.86 | -0.28 | 66.81 | +0.23 | 66.42 | +0.03 | 61.94 | +0.17 |
| 2 | 72.08 | +0.65 | 70.55 | -0.66 | 72.16 | +0.28 | 69.99 | +0.22 | 62.00 | +0.12 |
| 4 | 76.38 | +0.22 | 74.61 | +0.80 | 74.38 | +0.88 | 73.85 | +0.42 | 75.64 | +0.26 |
| 8 | 78.27 | +0.31 | 76.83 | +0.65 | 74.96 | +1.98 | 76.11 | +1.04 | 78.63 | +0.13 |
| 16 | - | - | 78.11 | +0.56 | 74.26 | +4.00 | 77.91 | +0.61 | - | - |

**Table 3: Test accuracies with different widths.** $\mu$P has a higher accuracy compared with SP in models with large widths. The learning rate is set slightly small to enlarge the effect of infinite width. Missing points are owing to the too-long computational time. The results for ResNet18 (CIFAR10) and ResNet50 (CIFAR100) are in the Appendix.G.1.

decrease the accuracy. These results highlighted that the $\mu$P for second-order optimization enables feature learning even in infinite-width models.

**ResNet:** We evaluate the test accuracy for SP and $\mu$P with VGG19 on CIFAR100, ResNet18 on CIFAR100 and ResNet50 on ImageNet. The number of channels in the middle layers is scaled from the original models. Original models correspond to width $= 4$. HPs are fixed regardless of width. Table 3 indicates that the accuracy can be consistently improved by $\mu$P. Figure 4 shows the learning curves for SP and $\mu$P with different widths. In SP, the test loss for width $= 16$ (wider model) is not necessarily higher than that for width $= 1$ (narrower model). However, in $\mu$P, wider models consistently match or outperform narrower models. This is more evident in the early stages of training.

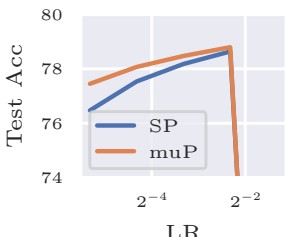

**Figure 5: $\mu$P consistently achieves higher accuracy for various learning rates.** (ResNet50 on ImageNet)

The advantage of $\mu$P generally holds regardless of a fixed learning rate. Figure 5 shows that regardless of the fixed learning rate, $\mu$P achieves higher test accuracy compared with SP. As a side note, the difference between SP and $\mu$P is particularly large when the learning rate is small.

## 5.2 LEARNING RATE TRANSFER

$\mu$P can transfer a learning rate for models with different widths. Learning rate transfer, as defined in a previous study, means that we can set the optimal learning rate to remain constant relative to the order of width, and that "wider is better" holds at this optimal learning rate. We confirm this across three different architectures, MLP, CNN, and ResNet in Figure 6. In the SP for SGD, the optimal learning rate decreases as the width increases. Similarly, when the damping heuristics of Equation (1) are used for SP in K-FAC, there is a shift in the optimal learning rate. These shifts in the optimal learning rate are caused by a lack of stability in these parameterizations. In $\mu$P, which is a stable parameterization, the optimal learning rate is fixed with respect to width. One can also confirm that the wider is better for a fixed HP. For instance, if MLP is trained with SP in K-FAC, test accuracy at width $= 16384$ is lower than width $= 512$. In contrast, if one uses $\mu$P for training, accuracy consistently increases with the width; thus, $\mu$P works as a better parameterization to promote feature learning.

## 5.3 DAMPING TRANSFER

The damping term is another HP that requires careful tuning in second-order optimization. In $\mu$P, we can re-use the damping obtained in a small model to the large-scale one. As shown in Figure 7, if one optimizes CNN with K-FAC using the damping heuristics (Eq. 14) in SP (i.e., the default setting), the optimal damping value increases as the width increases. In contrast, when damping is consistent with $\mu$P, it can be transferred across the widths.

## 6 CONCLUSION

We proposed a desirable parameterization inspired by $\mu$P for second-order optimization to achieve stable feature learning in infinite-width neural networks. As the computational resources are en-

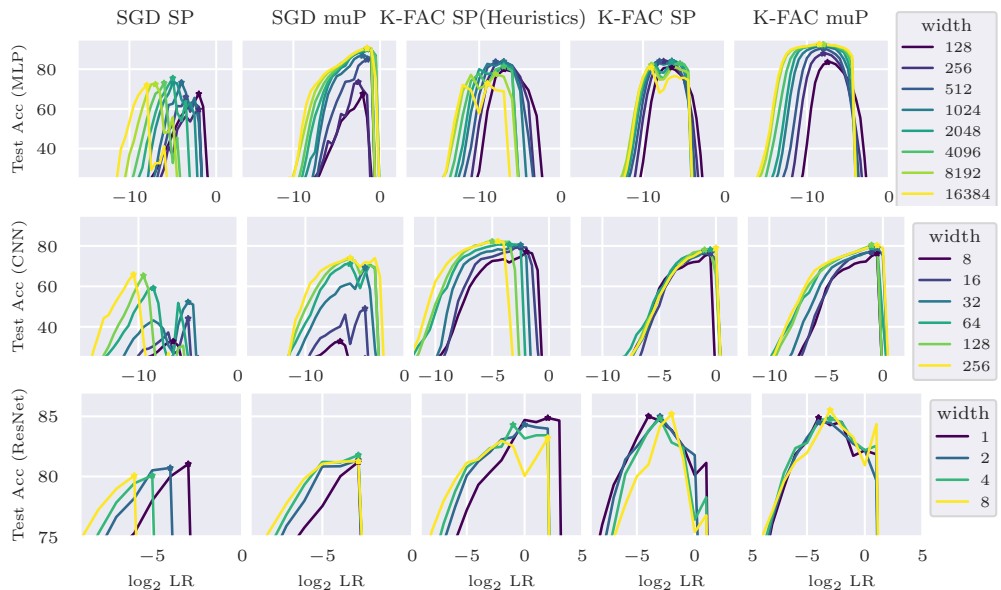

**Figure 6: $\mu$P allows the learning rate ($\eta'$) to transfer across widths.** Using $\mu$P, one can transfer the learning rate concerning width. In KFAC with SP, the heuristic damping obstructs the transfer of learning rates. With $\mu$P, the transfer succeeds and the wider models perform better They are trained by MSE loss with 1024 samples. This learning rate transfer also holds for the full dataset (Appendix.F.3). In addition, $\mu$P enables this transfer in Shampoo (Appendix.F.1) and FOOF (Appendix.F.2) as well.

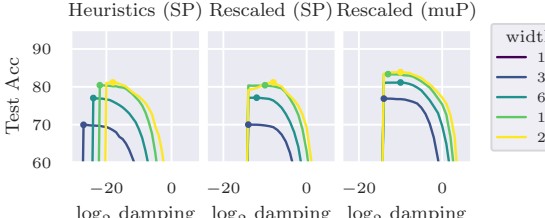

**Figure 7: Under the rescaled damping of K-FAC, the optimal damping is fixed regardless of width.** 3-layer CNN is trained on FashionMNIST by MSE loss (Full dataset size). The same transfer can also be observed for MLP (Appendix.H.2).

riched more and the model becomes larger, there is no guarantee that the current default settings of initialization, learning rate, and damping terms work well in a similar way to smaller models. In pursuit of such training of wider models, we empirically confirmed the effectiveness of the $\mu$P in commonly used second-order methods and the necessary modification of the HPs. It is also one of the advantages of the proposed parameterization that when using $\mu$P, HPs tuned in a smaller model can be transferred to a larger one.

**Limitation and future direction.** The current work and the framework of $\mu$P focus on the wide neural networks as large-scale models. It would be interesting to investigate the parameterization effective for the depth. Some studies have scaled weight initialization with respect to depth (Shoeybi et al., 2019; Radford et al., 2019). Transformer is a widely used architecture in modern deep learning; however second-order optimization has not yet been established well, and we have still limited empirical observation in both pertaining (Pauloski et al., 2022; Osawa et al., 2023b) and fine-tuning (Ding et al., 2023). Moreover, the finite width effect may be non-negligible in the Transformer; thus, we will need more careful study focused on it (Dinan et al., 2023; Wortsman et al., 2023). From a theoretical perspective, mathematically rigorous evaluation of feature learning in general steps is curious as well. Tensor Program (Yang & Littwin, 2023) is a strong candidate for such evaluation, although it is currently not applicable to the second-order parameterization and analytical evaluation is still limited even in the first-order gradient. We expect that our proposed parameterization and empirical observation will serve as a foundation for providing deeper insight into second-order optimization both in theory and practice.

ACKNOWLEDGMENTS

The authors thank Rio Yokota for his helpful comments and acknowledge the funding support from JST FOREST (Grant Number: JPMJFR226Q) and JSPS KAEKENHI (Grant Number: 22H05116, 23K16965).

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

# Appendices

## A  DEVIATION OF MUP

In this section, we first explain basic conditions and assumptions for $\mu$P in Section A.1. Next, we provide a derivation of $\mu$P for the first-order gradient in Section A.2. This deviation is based on a one-step update and its perturbation, which is easily generalized to the cases of K-FAC (in Section A.4.1) and Shampoo (in Section A.4.2).

### A.1  SETTING OF $\mu$P

Here, let us overview the conditions of $\mu$P proposed in Yang & Hu (2021). As we described in Section 3.2, the previous work defined the feature learning by

$$\Delta h_l := h_{l,t} - h_{l,0} = \Theta(1). \tag{S.1}$$

Note that in the appendices, we avoid using bold fonts of vectors and matrices in order to prevent a messy appearance. Intuitively speaking, the activation varies with the constant order to generate features in each layer. This is crucial for distinguishing the learning from the lazy regime (Jacot et al., 2018; Chizat et al., 2019). In the lazy regime, the model can converge to a global minimum but the trained parameters remain sufficiently close to their initial values and the model is equivalent to a linearized model. In other words, changes in the hidden layer are infinitesimal, i.e., $\Delta h_l \to 0$ while $\Delta f = \Theta(1)$ in the infinite width limit. Thus, the condition S.1 is required to avoid the lazy regime. For clarity, we set the width of the hidden layers to the same and take the infinite width limit [3]:

$$M_l = M, \quad M \to \infty \qquad (l = 1, ..., L-1). \tag{S.2}$$

To realize the feature learning, they proposed to use the abc-parameterization satisfying the following two conditions:

**Condition A.1** ($W_l$ updated maximally)**.**

$$\Delta W_{l,t} h_{l-1,t} = \Theta(1), \tag{S.3}$$

*where $\Delta W_{l,t} := W_{l,t} - W_{l,0}$.*

This first condition plays a fundamental role in feature learning. As is described in Section H.7 of Yang & Hu (2021) and we will explain more detail in the next subsection, this condition naturally appears when one considers the infinitesimal change of the parameter (i.e., $\partial_\eta$). The previous work also requires the following condition for $\mu$P:

**Condition A.2** ($W_L$ initialized maximally)**.**

$$W_{L,0} \Delta u_{L-1,t} = \Theta(1). \tag{S.4}$$

This condition plays a fundamental role for determining the variance of the weights in the last layer. Combining this condition with Condition A.1 makes the output change of order 1, that is, $f_t - f_0 = \Delta W_{L,t} h_{L-1,t} + W_{L,0} \Delta h_{L-1,t} = \Theta(1)$. The $\mu$P is derived from these two conditions. Subsequent studies have empirically verified that the successfully facilitates feature learning in various settings of actual training (Yang et al., 2021; Vyas et al., 2023).

Yang & Hu (2021) also assume

**Assumption A.3.** $u_{l,0}, h_{l,0} = \Theta(1)$ $(l < L)$, $\quad f_0 = u_{L,0} = O(1)$.

The input and output of the activation function are assumed to be of the order of 1. This assumption has been commonly employed in the training of neural networks (LeCun et al., 1998). Additionally,

---

[3]Our analysis remains applicable even if the widths of different layers vary, as long as all widths approach infinity in the same order. More precisely, even if the network width is given by $M_l = \alpha_l M$, with each $\alpha_l > 0$ being a constant, the $\mu$P remains the same in the infinite width limit of $M \to \infty$.

the $\mu$P setting allows that the network output $f$ can be close to zero at initialization. This assumption immediately leads to

$$a_1 + b_1 = 0, \tag{S.5}$$

$$a_l + b_l = 1/2 \quad (1 < l < L), \tag{S.6}$$

$$a_L + b_L \geq 1/2, \tag{S.7}$$

as is described in Theorem H.6 of Yang & Hu (2021). We also have

**Assumption A.4.** For feature learning, we suppose $a_L + b_L > 1/2$.

This is the Assumption H.23 of Yang & Hu (2021). This assumption is used to avoid technical subtleties potentially caused by $a_L + b_L = 1/2$: for random initialization, $f_0$ goes to 0 for $a_L + b_L > 1/2$ while it becomes Gaussian process with a non-zero variance for $a_L + b_L = 1/2$. Roughly speaking, this assumption ensures that a backpropagated error converges to a deterministic constant $(y - f_0) \to y$ in the infinite width limit and makes the derivation more clear by avoiding potential correlation between the error and trained features.

## A.2 $\mu$P's CONDITIONS OF ONE-STEP UPDATE AND PERTURBATION

As a preparation for the second-order optimization, let us explain the $\mu$P's conditions written in an infinitesimal one-step update. The essentially same perturbation is argued in Section H.7 of Yang & Hu (2021), but their explicit evaluation focuses on the development of feature kernels, i.e., $||h_{l,1}||^2$, and the precise evaluation of this value requires much-complicated argument. Since the current work is interested only in a simpler problem on the order evaluation of $h_{l,1}$, it will be more informative and clearer to show the perturbation argument specific to $h_{l,1}$ directly. As you can see below, this clarifies an explicit connection between feature learning and Conditions A.1 & A.2.

Express the first one-step update of the weight by

$$\Delta W_{l,1} = \frac{\eta'}{M^{2a_l + c_l}} G_l, \tag{S.8}$$

$$G_l := P_A \nabla_{w_l} \mathcal{L}_0 P_B, \tag{S.9}$$

where $P_A$ and $P_B$ are preconditioners. Note that $\nabla_{w_l} \mathcal{L}_0$ depends only on the initialization ($t = 0$). We have

$$W_{l,1} h_{l-1,1} = (W_{l,0} + \frac{\eta'}{M^{2a_l + c_l}} G_l) \phi(u_{l-1,1}(\eta)), \tag{S.10}$$

where we remarked the dependence of $\{W_{1,1}, ..., W_{l-1,1}\}$ on $\eta$ by $u_{l-1,1}(\eta)$.

If the derivative of the one-step update with respect to the learning rate is $\Theta(1)$, this implies that feature learning appears. While Yang & Hu (2021) explicitly evaluated $\partial_{\eta'} ||h_{l,1}||^2 \big|_{\eta'=0}$, let us evaluate here the following quantities which have an explicit expansion by the chain rule with respect to $\eta'$:

$$\partial_{\eta'} h_{l,1} \big|_{\eta'=0} = \phi'(u_l) \circ \frac{1}{M^{2a_l + c_l}} G_l h_{l-1,0} + \sum_{k=1}^{l-1} \frac{\partial h_{l,0}}{\partial u_{l-k}} \circ v(l,k) \qquad (l < L), \tag{S.11}$$

$$\partial_{\eta'} f_{t=1} \big|_{\eta'=0} = \frac{1}{M^{2a_L + c_L}} G_L h_{L-1,0} + W_{L,0} \partial_{\eta'} h_{L-1,1} \big|_{\eta'=0}, \tag{S.12}$$

where

$$v(l,k) := \frac{1}{M^{2a_{l-k} + c_{l-k}}} G_{l-k} h_{l-k-1,0}. \tag{S.13}$$

We can find connections between (S.11, S.12) and Conditions (A.1, A.2) as follows:

$$\partial_{\eta'} (\Delta W_{l,1} h_{l-1,1}) \big|_{\eta'=0} = \frac{1}{M^{2a_l + c_l}} G_l h_{l-1,0} \quad (l = 1, ..., L), \tag{S.14}$$

and

$$\partial_{\eta'} (W_{L,0} \Delta h_{L-1,1}) \big|_{\eta'=0} = W_{L,0} \partial_{\eta'} h_{L-1,1} \big|_{\eta'=0}. \tag{S.15}$$

That is, under the perturbation with a small $\eta'$, the conditions reduce to

**Condition A.1'**

$$\frac{1}{M^{2a_l+c_l}}G_l h_{l-1,0} = \Theta(1).$$ (S.16)

**Condition A.2'**

$$W_{L,0}\partial_{\eta'} h_{L-1,1}\big|_{\eta'=0} = \Theta(1).$$ (S.17)

After all, one can see that the Conditions of $\mu$P (A.1,A.2) reduces to explicit representations (S.16,S.17) under the perturbation. If Conditions A.1' and A.2' hold, it implies (S.11, S.12) of $\Theta(1)$ and then the feature learning (S.1) appears. One can say that the $\mu$P is the "maximal" update because it induces all layers trained as much as possible.

## A.3 CASE OF FIRST-ORDER OPTIMIZATION

As an exercise for preparing second-order optimization, we first evaluate Conditions A.1' and A.2' for the first-order gradient with the preconditioners $P_A = P_B = I$. The second-order cases are shown later.

**On Condition A.1'.** To avoid cumbersome notation, we omit the initialization index $t = 0$ as long as it doesn't lead to confusion. We can express the gradient $G_l$ by

$$G_l = \delta_l \text{diag}(\chi) h_{l-1}^\top,$$ (S.18)

where forward and backward signals are given by matrix forms $h_l, \delta_l \in \mathbb{R}^{M_l \times n}$. Note that the backward signals are given by the chain rule:

$$\delta_l = \phi'(u_l) \circ (W_{l+1}^\top \delta_{l+1}) \quad (l < L-1),$$ (S.19)
$$\delta_L = e_n,$$ (S.20)

where we defined $e_n := (1, ..., 1) \in \mathbb{R}^n$. We also introduced an error vector

$$\chi := y - f \in \mathbb{R}^n.$$ (S.21)

As is pointed out in the previous works, the important point of $G_l h_{l-1}$ (in other words, $\Delta W_l h_{l-1}$) is that $G_l$ and $h_{l-1}$ are not independent. We have

$$G_l h_{l-1,0} = \delta_l \text{diag}(\chi)(h_{l-1}^\top h_{l-1}).$$ (S.22)

Here, let us introduce

$$K_l^A := h_l^\top h_l / M, \quad K_l^B := \delta_l^\top \delta_l.$$ (S.23)

For the random initialization, these two matrices are well known as intermediate components of the neural tangent kernel, as discussed in Jacot et al. (2018); Yang (2020); Karakida & Osawa (2020). They are also known as essential components of the Fisher information matrix, which characterizes the geometric structure of the parameter space (Karakida et al., 2019; 2021). Assume that the activation function and its derivatives are polynomially-bounded. Then, in the infinite width limit, these two matrices become deterministic constants independent of the width. In more detail, the tensor program ensures almost sure convergence of the moments composed of the forward and backpropagated signals ($h_l$ and $\delta_l$) (Yang, 2020). The $l$-layer component of the neural tangent kernel is given by $K_A^l \circ K_B^l$ where $\circ$ denotes the Hadamard product. The kernel $(K_l^A)_{nn'}$ is $\Theta(1)$. Since we suppose the abc-parameterization, $\delta_{L-1} = \Theta(W_L) = \Theta(1/M^{a_L+b_L})$. This leads to

$$(K_l^B)_{nn'} = \Theta(1/M^{2(a_L+b_L)-1}),$$ (S.24)

for $1 \le l < L$, that is, the coordinate size is given by

$$\delta_l = \Theta(1/M^{(a_L+b_L)}).$$ (S.25)

After all, we have

$$\partial_{\eta'}(\Delta W_{l,1} h_{l-1,1})\big|_{\eta'=0} = \frac{1}{M^{2a_l+c_l-1}}\delta_l \text{diag}(\chi) A_{l-1} = \Theta(1/M^{r_l}),$$ (S.26)

with

$$
r_l = \begin{cases} 2a_1 + c_1 + (a_L + b_L) & (l = 1), \\ 2a_l + c_l - 1 + (a_L + b_L) & (1 < l < L), \\ 2a_L + c_L - 1 & (l = L), \end{cases} \tag{S.27}
$$

where we used the fact that $\chi$ is a constant vector of $\Theta(1)$ in the infinite width limit under Assumption A.4. Compared to $r_{1<l<L}$, $r_1$ does not include $-1$ because $M_0 = \Theta(1)$ and $A_0 = \Theta(1/M)$. The case of $r_L$ does not include $a_L + b_L$ because $\delta_L = \Theta(1)$.

**On Condition A.2'.** We can represent Eq. (S.17) by

$$
\partial_{\eta'}(W_{L,0}\Delta h_{L-1,1})
$$
$$
= W_{L,0}\left(\phi'(u_{L-1}) \circ (\partial_{\eta'}(W_{L-1,0}\Delta h_{L-2,1}) + \partial_{\eta'}(\Delta W_{L-1,1} h_{L-2,1}))\right). \tag{S.28}
$$

Note that

$$
W_{L,0}(\phi'(u_{L-1}) \circ \partial_{\eta'}(\Delta W_{L-1,1} h_{L-2,1})\big|_{\eta'=0})
$$
$$
= e_M(\delta_{L-1} \circ \frac{1}{M^{2a_{L-1}+c_{L-1}}} G_{L-1}h_{L-2}) \tag{S.29}
$$
$$
= \frac{1}{M^{2a_{L-1}+c_{L-1}-1}} e_M(\delta_{L-1} \circ (\delta_{L-1}\mathrm{diag}(\chi)A_{L-2})). \tag{S.30}
$$

Because $\delta_{L-1}$ has the coordinate size of Eq. (S.25) and the product with $e_M$ means the summation over $M$, we obtain

$$
\partial_{\eta'}(W_{L,0}\Delta h_{L-1,1})\big|_{\eta'=0} = \Theta(1/M^{a_L+b_L-1+r_{L-1}}). \tag{S.31}
$$

Finally, from Eqs. (S.27) and (S.31), the $\mu$P is given by

$$
\begin{cases} 2a_1 + c_1 + (a_L + b_L) = 0, \\ 2a_l + c_l - 1 + (a_L + b_L) = 0, \\ 2a_L + c_L - 1 = 0, \\ a_L + b_L - 1 = 0. \end{cases} \tag{S.32}
$$

## A.4  CASE OF SECOND-ORDER OPTIMIZATION

What we need to do here is to find the abc-parameterization satisfying Conditions A.1' and A.2' for the preconditioners $P_A$ and $P_B$ (S.9) of the second-order optimization methods. We assume the following:

**Assumption A.5.** The gradient of the loss $\nabla_\theta \mathcal{L}$, preconditioners $P_A$ and $P_B$ are calculated on the same batch of training samples at the first one step.

Furthermore, we consider the situation where the second-order optimization is valid, defined as follows.

**Definition A.6** (Valid second-order optimization). Suppose that each preconditioning matrix $X$ has a damping term $\rho_X$ as $X + \rho_X I$. We define the second-order optimization as *valid* if the eigenvalues of $X$ have an equal width-dependent scale to the damping term, that is, $\rho_X = \Theta(\|X\|_2)$.

This valid situation is rational because it prevents the effect of preconditioner from vanishing in the infinite width limit. It also avoids the damping term's faster convergence to zero, a scenario that can potentially lead to numerical instability when computing inverses for rank-deficient preconditioners.

### A.4.1  $\mu$P FOR K-FAC

**On Condition A.1':** For K-FAC, we have

$$
\frac{1}{M^{2a_l+c_l}} G_l h_{l-1} = \frac{1}{M^{2a_l+c_l}} (B_l + \rho_{B_l}I)^{-e_B} \delta_l \mathrm{diag}(\chi) h_{l-1}^\top (A_{l-1} + \rho_{A_{l-1}}I)^{-e_A} h_{l-1}, \tag{S.33}
$$

where

$$
A_l := h_l h_l^\top, \quad B_l := \delta_l \delta_l^\top. \tag{S.34}
$$

As a technical remark, we define (or implement) the metric matrix $B_l$ using $\delta_l = \nabla_{u_l} f_\theta$ for our derivations. If the implementation is based on the derivative on $w_l$, we have to multiply $1/M^{a_l}$ to $B_l$. Since the results are essentially the same, we are using more readable notations here without $a_l$.

Under Assumption A.5, for $e_A, e_B \in \{0, 1\}$, we can apply the push-through identity, i.e., $(I + XY)^{-1}X = X(I + YX)^{-1}$, as follows.

**(i) For $1 < l < L$**, We have

$$\frac{1}{M^{2a_l+c_l}}G_l h_{l-1} \tag{S.35}$$

$$= \frac{M^{e_B(2(a_L+b_L)-1)-e_A}}{M^{2a_l+c_l-1}}\delta_l \underbrace{(K_l^B + \rho_{B_l}M^{1-2(a_L+b_L)}I)^{-e_B}\text{diag}(\chi)(K_{l-1}^A + \rho_{A_{l-1}}MI)^{-e_A}A_{l-1}}_{=:K_l}. \tag{S.36}$$

It is noteworthy that in a similar way to Eq. (S.26), $K_l$ converges to a deterministic value in the infinite width limit. Since $K_l^A$ is a deterministic constant of $\Theta(1)$, its eigenvalues are also $\Theta(1)$. This means that the damping where the second-order optimization becomes valid is $d_{A_l} = -1$. As a side note, for $d_{A_l} < -1$, the damping term in $(K_l^A + \rho_{A_{l-1}}MI)^{-e_A}$ becomes dominant and the contribution of the preconditioner vanishes in the infinite width limit. In contrast, we may take $d_{A_l} > -1$ if $K_l^A$ is positive-definite. Similarly, we have $d_{B_l} = 2(a_L + b_L) - 1$.

we have $\partial_{\eta'}(\Delta W_{l,1}h_{l-1,1})\big|_{\eta'} = \Theta(1/M^{r_l})$ with

$$r_l = 2a_l + c_l + e_A + e_B - 1 - (2e_B - 1)(a_L + b_L) = 0. \tag{S.37}$$

**(ii) For $l = L$**, $\delta_L$ has the coordinate size $\Theta(1)$ and this means $\delta_L(\delta_L^\top\delta_L + \rho_B I)^{-e_B} = \Theta(1)$. Therefore, we easily obtain $d_{A_l} = -1$, $d_{B_L} = 0$ and

$$r_L = 2a_L + c_L + e_A - 1 = 0. \tag{S.38}$$

**(iii) For $l = 1$**, note that the input $h_0$ has the coordinate size $\Theta(1)$ and $(h_0^\top h_0 + \rho_A I)^{-e_A}h_0^\top = \Theta(1)$. Therefore, we have $d_{A_l} = -1$, $d_{B_L} = 2(a_L + b_L) - 1$ and

$$r_1 = 2a_1 + c_1 + e_B - (2e_B - 1)(a_L + b_L) = 0. \tag{S.39}$$

**On Condition 2'.** We have

$$W_{L,0}(\phi'(u_{L-1}) \circ \partial_{\eta'}(\Delta W_{L-1,1}h_{L-2,1}))\big|_{\eta'=0}$$

$$= e_M(\delta_{L-1} \circ \frac{1}{M^{2a_L+c_L}}G_{L-1}h_{L-2}) \tag{S.40}$$

$$= \frac{M^{e_B(2(a_L+b_L)-1)-e_A}}{M^{2a_l+c_l-1}}e_M(\delta_{L-1} \circ (\delta_{L-1}K_{L-1})), \tag{S.41}$$

with $K_l$ given by Eq. (S.36). To satisfy Condition 2', we need

$$a_L + b_L - 1 + r_{L-1} = 0. \tag{S.42}$$

Finally, by combining Conditions A.1' and A.2', we obtain the abc-parameterization of $\mu$P as

$$\begin{cases} 2a_1 + c_1 + e_B - (2e_B - 1)(a_L + b_L) = 0, \\ 2a_l + c_l + e_A + e_B - 1 - (2e_B - 1)(a_L + b_L) = 0, \\ 2a_L + c_L + e_A - 1 = 0, \\ a_L + b_L - 1 = 0. \end{cases} \tag{S.43}$$

By setting $a_l = 0$, we can eliminate the indeterminacy of the parameterization and obtain Proposition 4.1. By setting $e_A = e_B = 0$, we recover the case of the first-order gradient method.

### A.4.2 $\mu$P FOR SHAMPOO

Here, let us consider a general $e > 0$.

**On Condition A.1':** For Shampoo, we have

$$\frac{1}{M^{2a_l+c_l}}G_l h_{l-1} = \frac{1}{M^{2a_l+c_l}}(L_l + \rho_{L_l}I)^{-e/2}\delta_l\mathrm{diag}(\chi)h_{l-1}^\top(R_{l-1} + \rho_{R_{l-1}}I)^{-e/2}h_{l-1}, \quad \text{(S.44)}$$

where we use $\delta_l$ defined in Eq. (S.19).

**(i) For $1 < l < L$,** In a similar way to the case of K-FAC, we use the push-through identity. The update is given by

$$\frac{1}{M^{2a_l+c_l}}G_l h_{l-1} = \frac{1}{M^{2a_l+c_l}}(\delta_l\mathrm{diag}(\chi)h_{l-1}^\top h_{l-1}\mathrm{diag}(\chi)\delta_l^\top + \rho_{L_l}I)^{-e/2}\delta_l\mathrm{diag}(\chi)$$
$$\cdot h_{l-1}^\top(h_{l-1}\mathrm{diag}(\chi)\delta_l^\top\delta_l\mathrm{diag}(\chi)h_{l-1}^\top + \rho_{R_{l-1}}I)^{-e/2}h_{l-1}. \quad \text{(S.45)}$$

Note that we may multiply $1/M^{2a_l}$ to the metric matrices $L_l$ and $R_l$ when their implementation is based on the derivative on $w_l$.

Under Assumption A.5, we use the following push-through identity here:

**Lemma A.7** (e.g., Petersen et al. (2008)). *For any analytic function $g$ and real matrices $X$ and $Y$,*

$$g(XY)X = Xg(YX). \quad \text{(S.46)}$$

First, let us consider

$$Q := (h_{l-1}\mathrm{diag}(\chi)\delta_l^\top\delta_l\mathrm{diag}(\chi)h_{l-1}^\top + \rho_{R_{l-1}}I)^{-e/2}h_{l-1}. \quad \text{(S.47)}$$

Let us express the largest eigenvalue of $X$ by $\|X\|_2$. we have

$$\|h_{l-1}\mathrm{diag}(\chi)\delta_l^\top\delta_l\mathrm{diag}(\chi)h_{l-1}^\top\|_2 = \|\mathrm{diag}(\chi)\delta_l^\top\delta_l\mathrm{diag}(\chi)h_{l-1}^\top h_{l-1}\|_2 \quad \text{(S.48)}$$
$$= \Theta(1/M^{2(a_L+b_L)-2}). \quad \text{(S.49)}$$

By setting $\rho_{R_{l-1}} = (a_L + b_L) - 1$ and taking a certain constant $\rho'_{R_{l-1}}$ satisfying $\|h_{l-1}\mathrm{diag}(\chi)\delta_l^\top\delta_l\mathrm{diag}(\chi)h_{l-1}^\top\|_2/\rho_{R_{l-1}} < 1$, the inverse matrix in $Q$ has a matrix series expansion that converges. Thus, we can use Lemma A.7 and obtain

$$Q = h_{l-1}(\mathrm{diag}(\chi)\delta_l^\top\delta_l\mathrm{diag}(\chi)h_{l-1}^\top h_{l-1} + \rho_{R_{l-1}}I)^{-e/2}. \quad \text{(S.50)}$$

Applying the same argument to the $L_l$ side of (S.45), We obtain

$$\frac{1}{M^{2a_l+c_l}}G_l h_{l-1} = \frac{1}{M^{2a_l+c_l}}\delta_l(\mathrm{diag}(\chi)h_{l-1}^\top h_{l-1}\mathrm{diag}(\chi)\delta_l^\top\delta_l + \rho_{L_l}I)^{-e/2}\mathrm{diag}(\chi)$$
$$\cdot h_{l-1}^\top h_{l-1}(\mathrm{diag}(\chi)\delta_l^\top\delta_l\mathrm{diag}(\chi)h_{l-1}^\top h_{l-1} + \rho_{R_{l-1}}I)^{-e/2}. \quad \text{(S.51)}$$

In a similar way to Eq. (S.36), we obtain

$$r_l = 2a_l + c_l + 2e - 1 - (2e - 1)(a_L + b_L). \quad \text{(S.52)}$$

**(ii) For $l = 1, L$,** Similar to the case with K-FAC, by carefully handling $R_L$ for $l = L$ and $L_0$ for $l = 0$, we immediately find

$$r_1 = 2a_1 + c_1 + e - (2e - 1)(a_L + b_L), \quad \text{(S.53)}$$
$$r_L = 2a_L + c_L + e - 1. \quad \text{(S.54)}$$

**On Condition 2'.** In the same way as in the case of K-FAC, we have

$$a_L + b_L - 1 + r_{L-1} = 0. \quad \text{(S.55)}$$

In summary, we obtain the abc-parameterization of $\mu$P as

$$\begin{cases} 2a_1 + c_1 + e - (2e - 1)(a_L + b_L) = 0, \\ 2a_l + c_l + 2e - 1 - (2e - 1)(a_L + b_L) = 0, \\ 2a_L + c_L + e - 1 = 0, \\ a_L + b_L - 1 = 0. \end{cases} \quad \text{(S.56)}$$

### A.4.3 GAUSS-NEWTON METHOD

From the standpoint of computational cost, practical applications use approximate second-order optimization methods. For K-FAC and Shampoo two smaller pre-conditioners are applied to the first-order gradient in matrix form. Although other forms of second-order optimization are not so often utilized in deep learning, mentioning methods beyond K-FAC and Shampoo could be valuable, particularly from the perspective of demonstrating the theoretical applicability of these techniques.

Let us consider Gauss-Newton method in a layer-wise manner. We choose this setting because the K-FAC and Shampoo are layer-wise and it makes the comparison easier. It is also straightforward to apply similar order evaluation to the case without layer-wise approximation. The vector form of the Gauss-Newton gradient is given by

$$(\nabla_{\vec{w}_l}\mathcal{L}^\top \nabla_{\vec{w}_l}\mathcal{L} + \rho_l I)^{-1}\nabla_{\vec{w}_l}\mathcal{L} = (J_l^\top \mathrm{diag}(\chi^2)J_l + \rho_l I)^{-1}J_l^\top \chi \tag{S.57}$$

$$= J_l^\top (\mathrm{diag}(\chi^2)J_l J_l^\top + \rho_l I)^{-1}\chi, \tag{S.58}$$

where $J_l = \nabla_{\vec{w}_l} f$ is a $n \times M_l M_{l-1}$ Jacobian matrix. Put $\tilde{\chi}_l := (\mathrm{diag}(\chi^2)J_l J_l^\top + \rho_l I)^{-1}\chi$. Then, the matrix form of the gradient (S.58) is given by

$$\nabla_{w_l}\mathcal{L} = \delta_l \mathrm{diag}(\tilde{\chi}_l)h_{l-1}^\top, \tag{S.59}$$

under $a_l = 0$. This corresponds to $G_l$ in Eq.(S.18) and we can easily apply the same order evaluation shown in Section . The point is that $J_l J_l^\top$ is the NTK matrix for the $l$-th layer and we have

$$J_l J_l^\top = (\delta_l \delta_l^\top) \circ (h_{l-1}h_{l-1}^\top). \tag{S.60}$$

From Eq. (S.25) and $a_L + b_L = 1$, we can see that $J_l J_l^\top$ is $\Theta(1)$ for $1 < l < L$, $\Theta(1/M)$ for $l = 1$ and $\Theta(M)$ for $l = L$ and determines the order of $\tilde{\chi}$. Thus, the $\mu$P for the layer-wise Gauss-Newton method is as follows:

$$b_l = \begin{cases} 0 & l = 1 \\ 1/2 & 1 < l < L \\ 1 & l = L \end{cases}, \qquad c_l = 0, \qquad d_l = \begin{cases} 1 & l = 1 \\ 0 & 1 < l < L \\ -1 & l = L \end{cases}. \tag{S.61}$$

where $\rho_l = \rho_l'/M^{d_l}$. Thus, $b_l$ and $c_l$ are the same as in K-FAC and $\rho_l$ is the same as in Shampoo.

## A.5 EXTENSION TO OTHER LOSSES

Until now, we have considered the MSE loss with a one-dimensional target. Extending these results to multi-dimensional target cases is straightforward. Suppose that the target sample $y_i$ is a $C$-dimensional vector, with $C = \Theta(1)$. For a multi-dimensional target, its gradient (S.18) is given by

$$G_l = \tilde{\delta}_l \mathrm{diag}(\chi)(1_c \otimes h_l^\top). \tag{S.62}$$

We defined $\tilde{\delta}_l = [\delta_l^{(1)}, ..., \delta_l^{(C)}]$, where $\delta_l^{(k)}$ denotes the backward signal from the $k$-th output unit, and

$$\chi = y - f \in \mathbb{R}^{nC}. \tag{S.63}$$

Since $(1_c \otimes h_l^\top)h_l = (1_c \otimes h_l^\top h_l)$, we can apply the same arguments as those shown in Section A.2. Consequently, we can easily obtain the same $\mu$P for multi-dimensional targets.

Similarly, we can obtain the same $\mu$P for the case of a $C$-class cross-entropy loss. For SGD and Shampoo, we just need to replace the error vector in the gradient (S.18) with

$$\chi = y - \sigma(f) \in \mathbb{R}^{nC}, \tag{S.64}$$

where $y$ is the one-hot encoded true label vector and $\sigma(f)$ denotes a softmax function with the input logits $f$. For each input sample $x_i$, the softmax is defined by $\sigma(f_k(x_i)) := \exp(f_k(x_i))/\sum_{k'}^C \exp(f_{k'}(x_i))$ for $i = 1, ..., n$ and $k = 1, ..., C$. Thus, from the same argument as for MSE loss with a multi-dimensional target, we can obtain the same $\mu$P.

For K-FAC, note that the Fisher information matrix is given with the following backward part:

$$B_l = \tilde{\delta}_l \Lambda \tilde{\delta}_l^\top, \tag{S.65}$$

where $\Lambda(\sigma)$ is an $nC \times nC$ block diagonal matrix which is composed of $C \times C$ block matrices; $\mathrm{diag}(\sigma(f(x_i))) - \sigma(f(x_i))\sigma(f(x_i))^\top$ $(i = 1, ..., n)$. Because $\Lambda$ depends only on $f$ and its size is independent of the width, its contribution amounts to merely a constant multiplication. In addition, the only difference from the MSE case lies in the backward part. We can easily employ the push-through identity as

$$(\tilde{\delta}_l \Lambda \tilde{\delta}_l^\top + \rho_{B_l})^{-1} \tilde{\delta}_l = \tilde{\delta}_l (\Lambda \tilde{\delta}_l^\top \tilde{\delta}_l + \rho_{B_l})^{-1}. \tag{S.66}$$

Therefore, we can obtain the same $\mu$P as those presented in Proposition 4.1.

### A.6 Damping Heuristics

When using damping heuristics in K-FAC, second-order optimization does not become valid. In addition, $\Delta h_l$ decay with width if $a, b, c$ are set to $\mu$P settings. This mechanism can be explained as follows. Consider the damping of the input layer determined by Eq. (14).

$$\rho_{A_{l-1}}(= 1/\rho_{B_l}) := \sqrt{\frac{M}{\mathrm{tr}(B_l)}}\rho = \Theta(M^{(a_L+b_L)-1/2}). \tag{S.67}$$

According to Eq. (8), the appropriate damping scales are $\rho_A = \Theta(1)$ and $\rho_B = \Theta(1/M^{2(a_L+b_L)-1})$. Thus, if we use damping heuristics, second-order optimization is not valid because $\rho_A$ and $\rho_B$ are both larger than the appropriate damping scale. Since the order of the damping term is larger than the appropriate damping scaling and $\rho_A \cdot \rho_B = 1$, $\Delta W_1 h_0$ is of the same order as SGD. Thus the scale of $\Delta W_1 h_0$ will be as follows:

$$\partial_{\eta'}(\Delta W_{1,1} h_{0,1})\big|_{\eta'=0} = \Theta(1/M^{2a_1+c_1+a_L+b_L}). \tag{S.68}$$

Thus, $\Delta h_1$ will decay when $c_1 = 0$ and $b_L \geq 0$.

On the other hand, in the case of Shampoo, even if we use the damping of heuristics (determined by a constant multiple of the maximum eigenvalue), second-order optimization becomes valid. This mechanism can be explained as follows. The following three matrices all share the largest eigenvalue.

$$L_l + \rho I = \delta_l h_{l-1}^\top h_{l-1} \delta_l^\top + \rho I, \tag{S.69}$$

$$R_l + \rho I = h_{l-1} \delta_l^\top \delta_l h_{l-1}^\top + \rho I, \tag{S.70}$$

$$K_l + \rho I = h_{l-1}^\top h_{l-1} \delta_l^\top \delta_l + \rho I. \tag{S.71}$$

Since the matrix size of $K_l$ is independent of width, the maximum and mean eigenvalues of $K_l$ are of the same order with respect to width. Thus, the maximum eigenvalues of $R_l$ and $L_l$ are of the same order as the mean eigenvalue of $K_l$.

## B Experimental details

### B.1 Models

We considered the following models for vision tasks.

- **MLP:** We considered a 3-layer multilayer perceptron (MLP) with ReLU activation. The MLP models do not include bias.
- **CNN:** We considered 3-layer CNN with ReLU activation. The models consist of a two-layer convolution and a linear layer. We trained with different hidden widths where the width represents the output dimension of the first layer. Max pooling is applied after the activation function.
- **Myrtle-5:** We considered Myrtle family (Shankar et al., 2020) as an example for CNN. We trained with different hidden widths where the width represents the output dimension of the first layer.
- **ResNet:** We considered ResNet18 and ResNet50 (He et al., 2016) with different hidden widths where widths represent the number of channels. We used the existing implementation[4] for training CIFAR10. In addition, we used models from Pytorch Image models (Wightman, 2019) for ImageNet training.

---

[4]https://github.com/uoguelph-mlrg/Cutout

In addition to these four models, we also considered CBOW as a language model.

- **CBOW:** We used CBOW model for the word embedding task (Mikolov et al., 2013). The CBOW model is a network consisting of an embedding layer that embeds words and a linear layer that predicts context words. The models do not include an activation function.

## B.2 Details of Figures

This section explains the experimental setting for each figure. In all experiments, we implemented second-order optimization based on the ASDL library (Osawa et al., 2023a). Note that $\rho$ in the below represents a proportionality constant for damping as $\rho_A^{Re} = \frac{\rho}{M_{l-1}} \operatorname{tr}(\boldsymbol{h}_{l-1}^\top \boldsymbol{h}_{l-1})$ and $\rho_B^{Re} = \frac{\rho}{M_l} \operatorname{tr}(\boldsymbol{\delta}_l^\top \boldsymbol{\delta}_l)$.

**Figure 1** In the upper graph, we trained a 3-layer MLP on the MNIST dataset with cross-entropy loss. In the second graph, we trained a Myrtle-5 on the CIFAR10 with cross-entropy loss. Both graphs represent $\Delta h_l$ at the 10th iteration. The training sets have been reduced to 256 samples, and we trained models by full-batch training. We apply no data augmentation.

**Figure 2** We trained a 3-layer MLP on FashionMNIST with $\eta = 0.001$, $\rho = 1$. Its parameterization follows $\mu$P settings. We apply no data augmentation.

**Figure 3** We trained a 3-layer CNN on FashionMNIST with different $b_L$ for 50 epochs. This graph shows the ratio between the gradient and the preconditioned gradient. Layer-wise learning rate follows the scaling of $c$ in muP in Eq. 9. The learning rate is tuned by grid search. Its settings are as follows.

- Learning rate : $2^z$ where $z \in \{1, 0, -1, ..., -20\}$
- Damping term $\rho$ for K-FAC: 1
- Damping term $\rho$ for Shampoo: 1e-3

In Table.2, we experimented with exactly the same settings as above, but with different batch sizes.

**Figure 4 (Left)** We trained Context as a Bag-of-Words (CBOW) on WikiText2 by Shampoo with $\eta = 0.1$, $\rho = 10^{-6}$ and $\tau = 0.1$. We test the embeddings by the word analogy task on WordSim353. We used words that appeared more than 128 times in the dataset.

**Figure 4 (Right)** We trained ResNet18 on CIFAR100 by K-FAC with $\eta = 0.003$, $\rho = 10$ and $\tau = 0.5$. We use a cross-entropy loss with label smoothing. We apply RandomCrop, RandomHorizontalFlip, AutoAugment, and Cutout as data augmentation. In Table 3, we experimented with exactly the same settings when training ResNet18 on CIFAR100.

**Figure 5** We trained ResNet50 on ImageNet by K-FAC with $\rho = 0.001$ and $\tau = 0.05$. We use a cross-entropy loss with label smoothing. We apply RandomCrop, RandomHorizontalFlip, and Cutout as data augmentation. In addition, to prevent instability in the training, we used gradient clipping in Grosse & Martens (2016). In Table 3, we experimented with exactly the same settings when training ResNet50 on ImageNet.

**Figure 6** In the first row, we trained 3-layer MLP on MNIST for 20 epochs. The number of samples is reduced to 1024 and trained by MSE loss. In training with K-FAC, we used $\rho = 0.001$ for heuristics damping and $\rho = 1$ for rescaled damping.

In the second row, we trained a 3-layer CNN on FashionMNIST for 50 epochs. The number of samples is reduced to 1024 and trained by MSE loss. In training with K-FAC, we used $\rho = 1$ for heuristics damping and $\rho = 100$ for rescaled damping.

In the last row, we trained ResNet18 on FashionMNIST for 20 epochs. The number of samples is reduced to 1024 and trained by MSE loss. In training with K-FAC, we used $\rho = 10$ for heuristics damping and $\rho = 10$ for rescaled damping.

**Figure 7** We trained a 3-layer CNN on FashionMNIST for 50 epochs with $\eta = 0.003$. The number of samples is reduced to 1024 and trained by MSE loss.

**Table 3** We trained VGG19 on CIFAR100 for 300 epochs, ResNet18 on CIFAR100 for 300 epochs and ResNet50 on ImageNet for 55 epochs. For VGG19 training with K-FAC, we set the learning rate $\eta = 0.01$, for ResNet18 training with shampoo, we set $\eta = 0.001$, and for ResNet50 training with K-FAC, we set $\eta = 0.2$. In all other settings, $\eta = 0.003$.

## C  ADDITIONAL DESCRIPTION OF SECOND-ORDER OPTIMIZATION

### C.1  K-FAC

Natural gradient descent (Amari, 1998) is an optimization algorithm that preconditions the gradient by the inverse of the Fisher information matrix. Its update rule is given by

$$\boldsymbol{\theta}_{t+1} = \boldsymbol{\theta}_t - \eta_t \left( \boldsymbol{F}(\boldsymbol{\theta}_t) + \rho I \right)^{-e} \nabla \mathcal{L}(\boldsymbol{\theta}_t), \tag{S.72}$$

where

$$\boldsymbol{F}(\boldsymbol{\theta}) = \mathbb{E}_{\boldsymbol{x} \sim q(\boldsymbol{x}), \boldsymbol{t}' \sim p_{\boldsymbol{\theta}}(\boldsymbol{t}'|\boldsymbol{x})} \left[ \nabla \log p_{\boldsymbol{\theta}} \left( \boldsymbol{t}' \mid \boldsymbol{x} \right) \nabla \log p_{\boldsymbol{\theta}} \left( \boldsymbol{t}' \mid \boldsymbol{x} \right)^\top \right], \tag{S.73}$$

is the Fisher information matrix. Here, we use the general exponent $e$. While $e = 1$ is commonly used, NGD with $e < 1$ has also been explored in some studies (Huh, 2020; Amari et al., 2021).

K-FAC (Martens & Grosse, 2015; Grosse & Martens, 2016) is an approximation algorithm for NGD. It approximates $\boldsymbol{F}_l(\boldsymbol{\theta}_t)$ by Kronecker product of two matrices

$$\boldsymbol{F}_l(\boldsymbol{\theta}_t) + \rho I \approx (B_l + \rho_B I) \otimes (A_{l-1} + \rho_A I), \tag{S.74}$$

where $B_l = \mathbb{E}\left[ \delta_l {\delta_l}^\top \right]$ and $A_l = \mathbb{E}\left[ h_l {h_l}^\top \right]$. Exponential moving average is often utilized to stabilize the estimation of the K-FAC curvature matrix.

$$A_l^{(t+1)} = \xi A_l^{(t)} + (1 - \xi) \mathbb{E}\left[ h_l {h_l}^\top \right], \tag{S.75}$$

$$B_l^{(t+1)} = \xi B_l^{(t)} + (1 - \xi) \mathbb{E}\left[ \delta_l {\delta_l}^\top \right]. \tag{S.76}$$

### C.2  SHAMPOO

Adaptive gradient descent utilizes the square root of batched empirical Fisher

$$\boldsymbol{F}_{\text{emp}}^{\text{batch}}(\boldsymbol{\theta}) = \mathbb{E}_{\mathcal{B} \sim p_{\text{data}}} \left[ \nabla \mathcal{L}_{\mathcal{B}}(\boldsymbol{\theta}) \nabla \mathcal{L}_{\mathcal{B}}(\boldsymbol{\theta})^T \right], \tag{S.77}$$

instead of $\boldsymbol{F}(\boldsymbol{\theta}_t)$ where $\nabla \mathcal{L}_{\mathcal{B}}(\boldsymbol{\theta}) = \frac{1}{|\mathcal{B}|} \sum_{(\boldsymbol{x}, \boldsymbol{t}) \in \mathcal{B}} \nabla \log p_{\boldsymbol{\theta}}(\boldsymbol{t} \mid \boldsymbol{x})$ represents the mini-batch gradient. Shampoo approximates this batched empirical Fisher by Kronecker product of two matrices (Gupta et al., 2018).

$$\boldsymbol{F}_{\text{emp}}^{\text{batch}}(\boldsymbol{\theta}) + \rho I \approx (R_l + \rho_R I) \otimes (L_{l-1} + \rho_L I), \tag{S.78}$$

where $L_l = \mathbb{E}[\delta_l {h_{l-1}}^\top h_{l-1} {\boldsymbol{\delta}_l}^\top]$ and $R_l = \mathbb{E}[h_{l-1} {\delta_l}^\top \delta_l {h_{l-1}}^\top]$. Summation is often utilized to stabilize the estimation of the Shampoo curvature matrix while exponential moving average is also well utilized.

$$L_l^{(t+1)} = L_l^{(t)} + \mathbb{E}[\delta_l {h_{l-1}}^\top h_{l-1} {\boldsymbol{\delta}_l}^\top], \tag{S.79}$$

$$R_l^{(t+1)} = R_l^{(t)} + \mathbb{E}[h_{l-1} {\delta_l}^\top \delta_l {h_{l-1}}^\top]. \tag{S.80}$$

This summation prevents $L_l^{(t)}$ and $R_l^{(t)}$ from becoming a zero matrix, which ensures stable learning even at an infinite width limit.

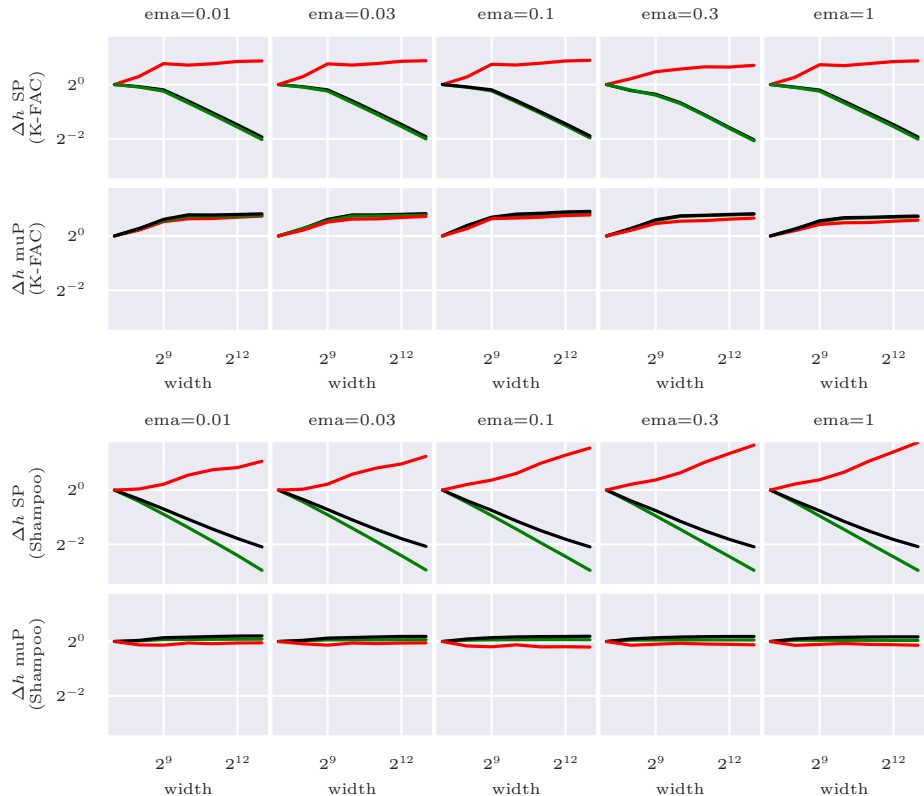

**Figure S.1 : Exponential moving average does not affect order evaluation.** Exponential moving average has almost no effect on the behavior of $\Delta h$. More precisely, regardless of the exponential moving average, $\Delta h_l$ decays in the input and hidden layers with increasing width when trained with SP, but it does not decay when trained with $\mu$P. We trained a 3-layer MLP on CIFAR10. In K-FAC we used rescaled damping.

## D  ADDITIONAL EXPERIMENTS ON EMPIRICAL VERIFICATION OF $\mu$P

### D.1  ADDITIONAL EXPERIMENTS ON $\Delta h$

**Exponential moving average**   If we take an exponential moving average over the curvature matrix, we cannot use the push-through identity in Eq. 10. However, even in this case, the behavior of $\Delta h$ is consistent with the order evaluation. Figure S.1 shows that the exponential moving average has almost no effect on the order of $\Delta \boldsymbol{h}$. This shows that even if we use an exponential moving average, $\mu$P is still reasonable.

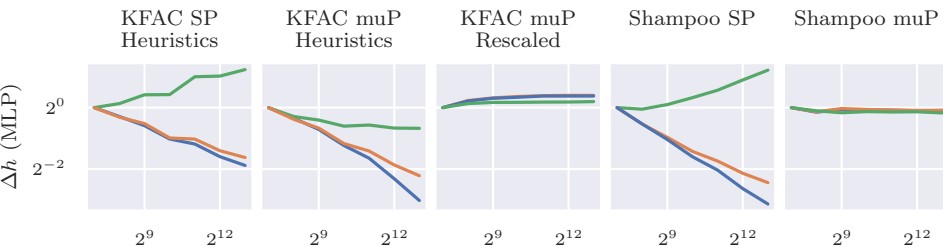

**Figure S.2 : $\mu$P achieves feature learning in MLP with tanh activation** We trained a 3-layer MLP on MNIST using cross-entropy loss. Even if the activation function for the MLP is tanh, we observed the same behavior as in Figure 1.

**Activation Function** The behavior of $\Delta \boldsymbol{h}$ is not affected by the activation function. We trained 3-layer MLP with tanh activation in Figure S.2 . It shows the same behavior as when we trained 3-layer MLP with relu activation (Figure 1).

# E ADDITIONAL EXPERIMENTS ON THE VARIANCE OF THE LAST LAYER

## E.1 TRAIN CURVE

Figure S.3 shows the learning curves for the experiment in Figure 3. K-FAC can reach the NNGP solution in one step. Thus, we can observe that K-FAC reaches high test accuracy from the first step for $b = 64$. However, the accuracy at $b = 1$ is higher than that at $b = 64$ since the training process for $b = 64$ stays at the initial value.

Note that the optimal learning rate of $b = 64$ is smaller than that of $b = 0.5$ or $b = 1$. This is because the NNGP solution is a sharp local solution, and a large learning rate makes the subsequent behavior unstable.

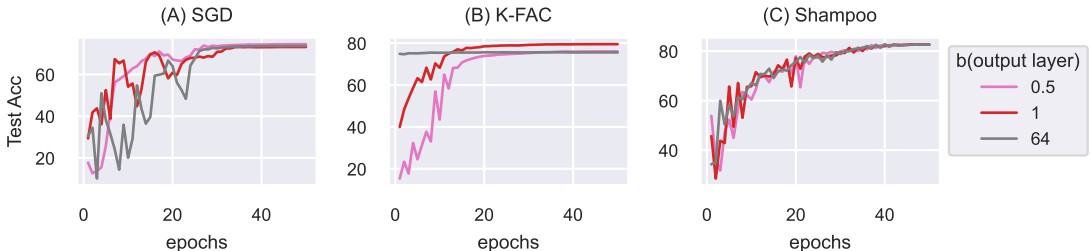

**Figure S.3 : Train Curve Comparison for different** $b_L$ K-FAC achieves high accuracy in one step when $b_L = 64$ since K-FAC can achieve NNGP solution in one step. However, K-FAC can not escape from the NNGP solution. In this figure, we trained 3-layer CNNs on Fashion-MNIST with batch size = 1024.

## E.2 CHOICE OF LOSS FUNCTION

The experiment in Figure 3 is performed with MSE loss to clarify the connection with NNGP solution. This result is easily extended from MSE loss to cross-entropy loss. Figure S.4 shows that even with a cross-entropy loss, K-FAC shows higher accuracy only when $b_L$ is at or near 1 in the last layer.

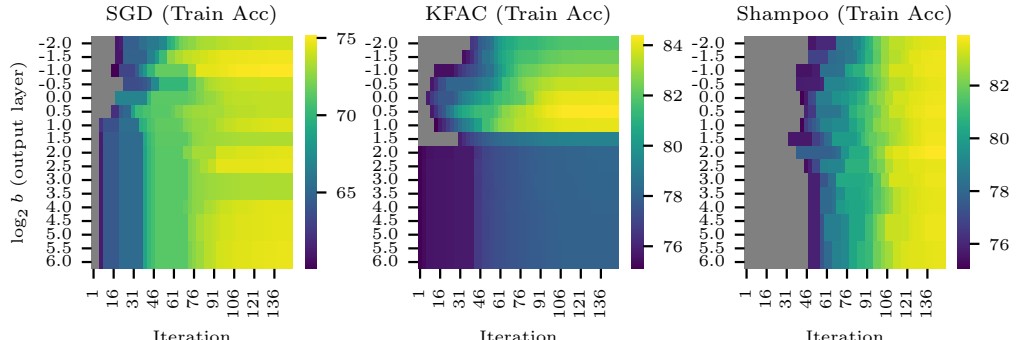

**Figure S.4 : K-FAC converges to NNGP solution even in the cross-entropy loss.** We experimented with a 3-layer CNN, moving only the last layer $b_L$ from $\mu$P. The loss function is cross-entropy. K-FAC shows that the regions of high accuracy are concentrated only around $b = 1$.

### E.3 Choice of model architecture

The experiment in Figure 3 was performed with a 3-layer CNN model. This result can be generalized to other architectures. Figure S.5 indicates that accuracy at $b_L = 1$ is higher than $b_L = 0.5$. In this example, train accuracy for $b_L \gg 1$ is consistently higher than the train accuracy for $b_L = 0.5$. In ResNet-18, the maximum value appears to be obtained at $b_L = 2$, but this may be due to the constant factor rather than order, which occurs when experiments are conducted with finite widths.

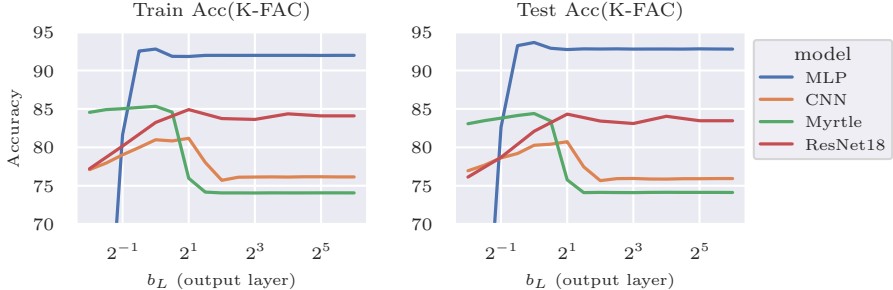

**Figure S.5 : $\mu$P is effective in various architecture.** According to the K-FAC $\mu$P, $b_L = 1$ has a higher accuracy than $b_L = 0.5$. This is consistent across various models. MLPs are trained with MNIST, while other models are trained with FashionMNIST. The number of samples is reduced to 1024. MLP and CNN are trained in full batches, while Myrtle and ResNet18 are trained in mini-batches with a batch size of 128.

### E.4 On the effect of Momentum

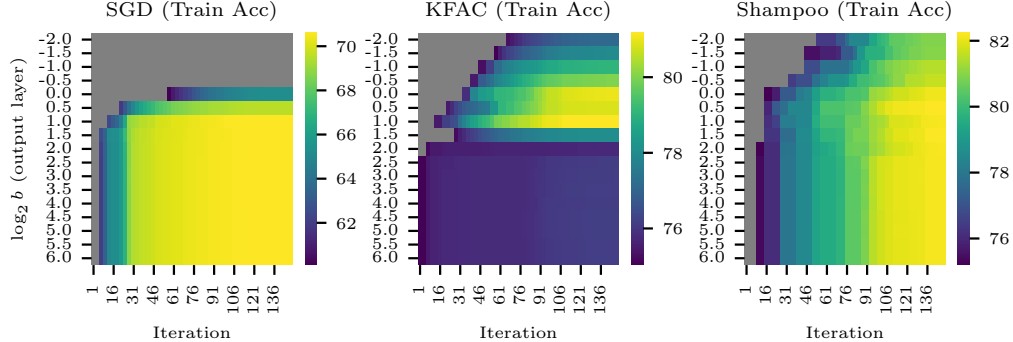

**Figure S.6 : K-FAC converges to NNGP solution even without momentum.** We experimented with a 3-layer CNN, moving only the last layer $b_L$ from $\mu$P. The loss function is cross-entropy. K-FAC shows that the regions of high accuracy are concentrated only around $b = 1$.

**Figure S.7 : $\mu$P is effective across batch-size without momentum** We trained FashionMNIST with a 3-layer CNN with different batch sizes. We did not use momentum in this figure. The values represent the accuracy of the training dataset.

| Optimizer | $b$ | Batch Size | | | | |
|---|---|---|---|---|---|---|
| | | 4 | 16 | 64 | 256 | 1024 |
| | 0.5 | 80.52 | 78.69 | 75.05 | 67.94 | 49.70 |
| SGD | 1.0 | 82.85 | 80.41 | **77.58** | 73.12 | 64.30 |
| | 64.0 | **83.59** | **81.23** | 77.25 | **73.63** | **70.53** |
| | 0.5 | 77.60 | 79.66 | **83.94** | 82.14 | 78.63 |
| K-FAC | 1.0 | **79.10** | **81.69** | 83.92 | **83.16** | **80.27** |
| | 64.0 | 76.06 | 76.95 | 77.28 | 76.04 | 75.94 |

Figure S.6 and Figure S.7 represents the results for training 3-layer CNN without momentum. Since momentum is known to have the effect of escaping from the saddle point, it is expected that it is more difficult to escape from the NNGP solution without momentum. However, there was almost no effect of removing momentum on the relationship between accuracy and $b_L$ while the overall accuracy is lowered by removing momentum.

### E.5 WORD2VEC

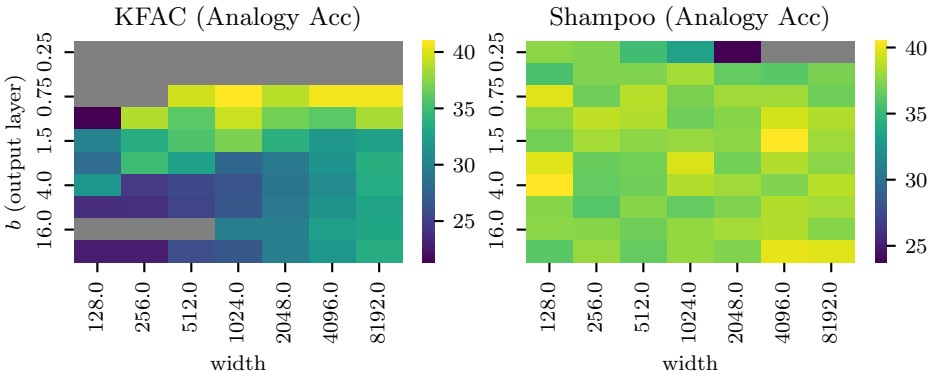

**Figure S.8 : Effect of $b_L$ on Accuracies of Word Analogy** In K-FAC, only a limited range of $b_L$ has high accuracy, whereas Shampoo has nearly 40% accuracy over a wide range of $b_L$. $c_l$ is correctly set to match the $\mu$P setting.

Feature learning is especially important in word2vec, which requires careful tuning of $b_L$ in K-FAC. Figure S.8 examines the effect of $b_L$ on accuracy in K-FAC and Shampoo. In this figure, HPs are tuned with the following settings. Note that we tuned the variance of weights at width=128 since K-FAC is sensitive to the variance of the last layer.

- learning rate = {1e-1, 1e-2, 1e-3 }
- embedding weight multiplier = {1, 1e-1, 1e-2}
- output weight multiplier = {1, 2, 4, 8, 16}

As shown in Figure S.8 , K-FAC achieves high accuracy only around $b_L = 1$. On the other hand, Shampoo achieves high accuracy in a wide range of $b_L$. This example implies that while K-FAC learns features in Word2Vec, K-FAC requires more careful tuning of $b_L$ than Shampoo.

## F ADDITIONAL EXPERIMENTS ON LEARNING RATE TRANSFER

### F.1 SHAMPOO

Figure S.9 shows that $\mu$P for Shampoo allows the learning rate tuned in a small model to be re-used in a larger model. In Shampoo, the learning rate is stable even in SP while in $\mu$P, the wider model achieves higher accuracy over a wide range of learning rates.

### F.2 FOOF

Since we introduced general exponents $e_A$ and $e_B$, we can consider $\mu$P for FOOF. The $\mu$P for FOOF is immediately derived from Proposition 9.

$$b_l = \begin{cases} 0 & l = 1 \\ 1/2 & 1 < l < L \\ 1 & l = L \end{cases}, \qquad c_l = \begin{cases} -1 & l = 1 \\ -1 & 1 < l < L \\ 0 & l = L \end{cases}, \qquad d_l^A = \begin{cases} -1 & 1 < l \le L \\ 0 & l = 1 \end{cases}.$$

$$(\text{S.81})$$

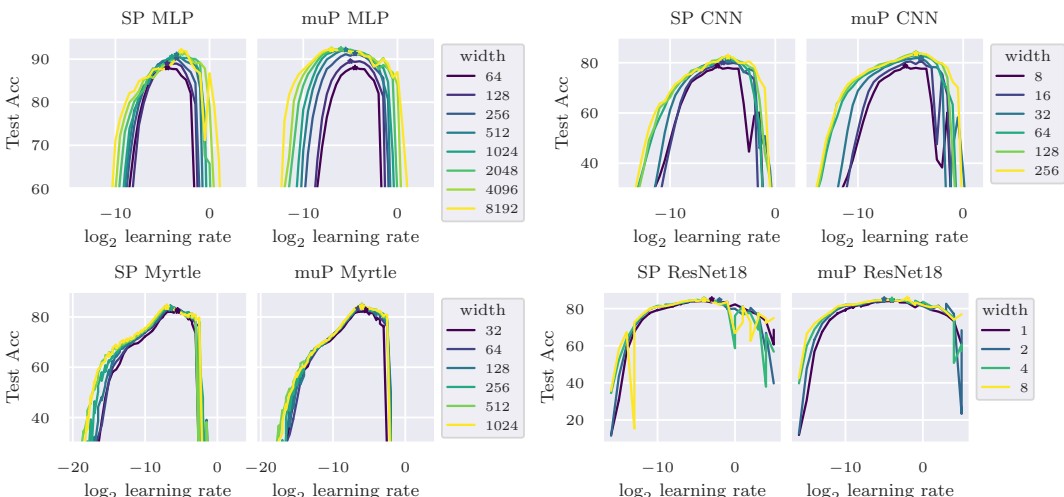

**Figure S.9 :** $\mu$**P for Shampoo allows the learning rate ($\eta'$) to transfer across widths.** We trained various models with SP and $\mu$P by Shampoo. We trained 3-layer MLP on MNIST, 3-layer CNN on FashionMNIST, Myrtle-5 on FashionMNIST and ResNet18 on FashionMNIST. The number of samples is reduced to 1024.

Figure S.10 shows that $\mu$P for FOOF enables the learning rate to transfer across the width as well as in SGD or K-FAC. The learning rate may be transferred across widths since the SP for FOOF is a stable parameterization. In addition, we can observe that accuracy obtained by training with $\mu$P is higher than SP, especially in wide models.

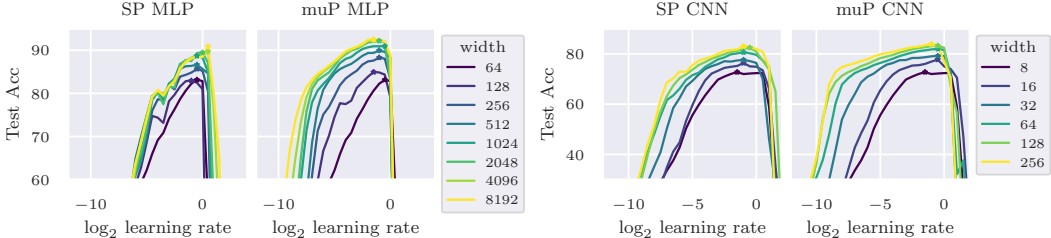

**Figure S.10 :** $\mu$**P for FOOF allows the learning rate ($\eta'$) to transfer across widths.** We trained a 3-layer MLP on MNIST and a 3-layer CNN on FashionMNIST. The number of samples is reduced to 1024.

### F.3 LEARNING RATE TRANSFER WHEN THE SAMPLE SIZE IS FULL

In Figure 6, we reduced the training samples. By reducing the dataset size, finite-width models are known to behave more closely to infinite-width models, as has often been seen in papers examining the theoretical aspects of second-order optimization and feature learning(Geiger et al., 2020; Karakida & Osawa, 2020; Amari et al., 2021). However, even when training on the full dataset, its learning rate can be transferred using $\mu$P as shown in Figure S.11 . In addition, we can observe that the learning rate is correctly transferred when we optimize ResNet50 on ImageNet. As shown in Figure S.1 , the optimal learning rate is fixed at 0.2 regardless of the width.

### F.4 ACTIVATION FUNCTION AND LOSS FUNCTION

In Figure 6, we trained 3-layer MLP with ReLU activation by MSE Loss. Similar observations can be made when using Tanh activation or cross-entropy loss as shown in Figure S.12 . Note that squashing activation functions such as tanh and softmax are not recommended in some experiments in Yang

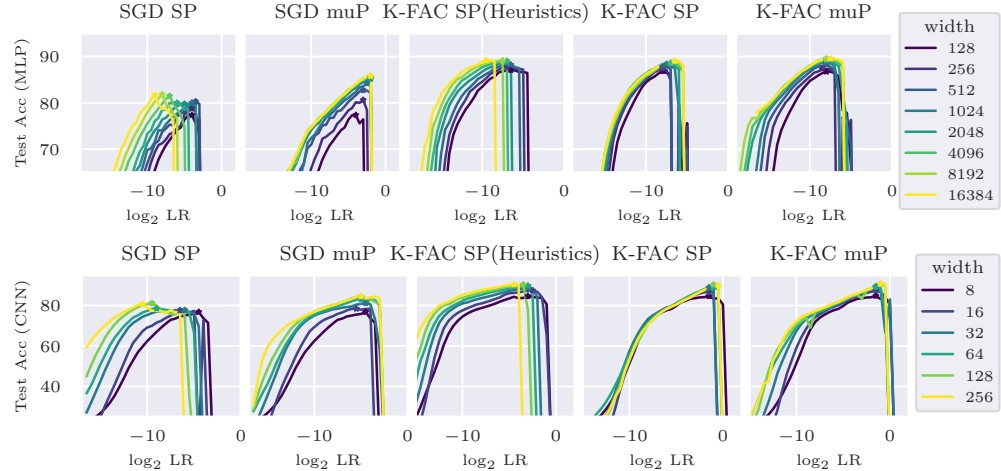

**Figure S.11 : $\mu$P allows the learning rate ($\eta'$) to transfer across widths(Full Dataset)**
Using $\mu$P, one can transfer the learning rate with respect to width. We trained a 3-layer MLP on FashionMNIST and a 3-layer CNN on FashionMNIST. In contrast to Figure 6, the sample size is not been reduced. In KFAC with SP, we use damping heuristics which prevent the transfer of learning rate.

|  | Learning Rate | | | | |
|---|---|---|---|---|---|
| width | 0.025 | 0.050 | 0.100 | 0.200 | 0.400 |
| 0.5 | 69.26 | 70.07 | 70.19 | **70.56** | 70.15 |
| 1.0 | 74.35 | 74.67 | 75.54 | **75.92** | 75.32 |
| 2.0 | 77.45 | 78.07 | 78.47 | **78.79** | 47.64 |

**Table S.1 : optimal learning rate does not shift in training ResNet50 on ImageNet.** The optimal learning rate is fixed at 0.2 and does not shift when scaling the width.

et al. (2021) D.3. However, the properties of the hyperparameter landscape for each parameterization in Figure S.12 are almost the same as in Figure 6.

# G  ADDITIONAL EXPERIMENTS ON TRAINING WIDE MODELS

## G.1  ANOTHER DATASET AND MODELS

In Figure 3, we compare the test accuracy with SP and $\mu$P and find $\mu$P takes a higher accuracy than SP in wide models. This tendency is also true for other settings. As shown in Figure S.13 and Figure S.14 , $\mu$P achieves higher valuation accuracy than SP in many cases. Specifically, the difference between SP and $\mu$P is larger for wider models, and the difference is often larger in the early stages of learning.

## G.2  ERROR BAR

Table.S.2 shows the results of the comparison of the accuracy of SP and $\mu$P with standard deviation. There is a significant difference in accuracy between SP and $\mu$P regardless of the effect of seed.

## G.3  DISTANCE FROM INITIAL WEIGHT

The relative distance from their initial weights value $\Delta W/|W|$ is often used in the analysis of infinite-width models. If $\Delta W/|W|$ is always zero throughout training, the model will fall into a lazy regime. As shown in Figure S.15 , in the early stages of training with SP, $\Delta W/|W|$ of wide-width models is almost zero. As the training proceeds, the weights begin to deviate from their initial values, which can be considered as finite-width effects. The wider the width, the longer the time that $\Delta W/|W|$ is almost zero. This suggests that the weights do not move away from their initial values in the infinite-width limit. In $\mu$P, however, the point at which weights start to move away from their initial

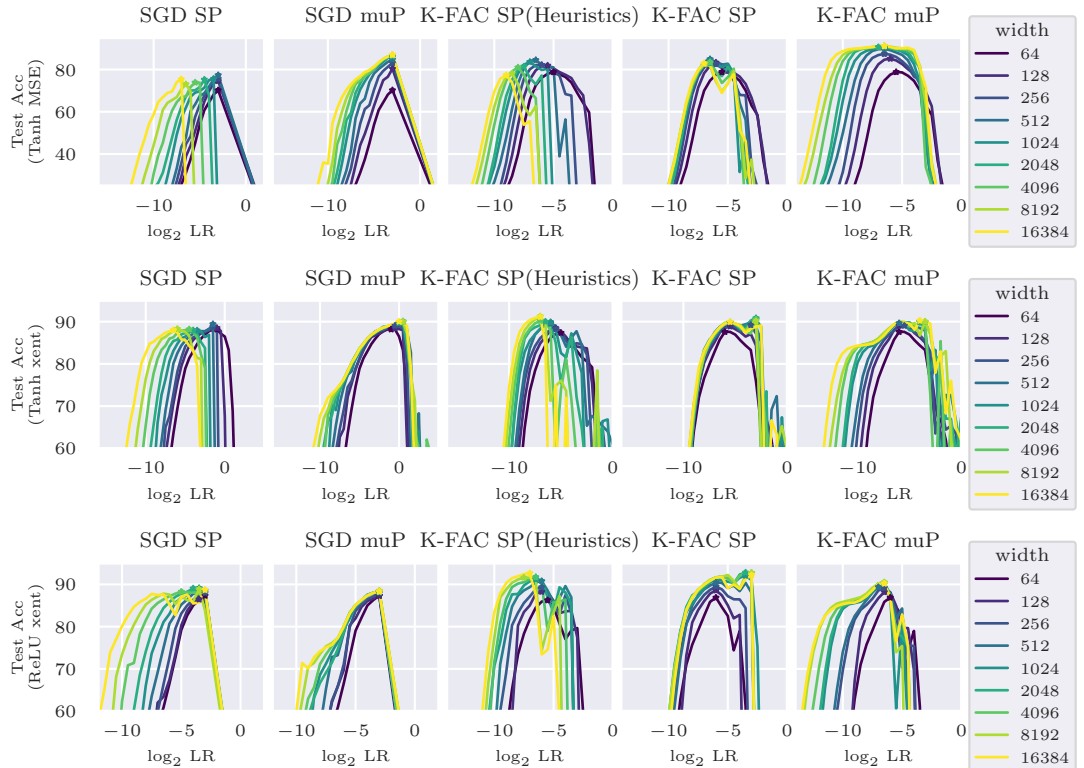

**Figure S.12 : Learning rate transfer across different activation functions and loss functions.**
We trained a 3-layer MLP on MNIST. We used tanh activation with MSE loss in the first row, tanh activation with cross-entropy loss in the second row, and ReLU activation with cross-entropy loss in the last row. We trained on MNIST dataset and the sample size was reduced to 1024.

| | Epochs=100 | | Epochs=200 | | Epochs=300 | |
| | K-FAC | Shampoo | K-FAC | Shampoo | K-FAC | Shampoo |
| wid | SP / $\mu$P | SP / $\mu$P | SP / $\mu$P | SP / $\mu$P | SP / $\mu$P | SP / $\mu$P |
|---|---|---|---|---|---|---|
| 1 | 89.54/+0.00 | 88.27/+0.43 | 91.35/+0.00 | 91.01/-0.25 | 91.97/+0.00 | 91.57/-0.14 |
| 2 | 92.10/+0.26 | 91.31/+0.40 | 93.92/+0.30 | 93.62/+0.14 | 94.32/+0.03 | 93.95/+0.05 |
| 4 | 93.64/+0.57 | 93.32/+0.22 | 95.11/+0.45 | 95.19/-0.15 | 95.57/+0.14 | 95.56/-0.10 |
| 8 | 93.56/+0.74 | 94.02/+0.33 | 95.44/+0.20 | 95.81/+0.19 | 95.73/+0.44 | 96.09/-0.04 |
| 16 | 93.00/+1.36 | 94.81/+0.96 | 95.04/+0.66 | 96.28/+0.14 | 95.31/+0.72 | 96.49/+0.24 |

**Figure S.13 : Test Accuracies of ResNet18 on CIFAR10.** We trained ResNet18 on CIFAR100 with $\eta = 0.003, \rho' = 10$. $\mu$P has a higher accuracy than SP in models with wide widths.

values remains the same, even as the width increases, which implies that training can proceed even at an infinite width limit.

## H    ADDITIONAL EXPERIMENTS ON DAMPING TRANSFER

### H.1    ON FIXED DAMPING SCALING

In this paper, we determine damping mainly by Eq. 15 as this is consistent with $\mu$P. However, even if damping is determined by Eq. 7, damping is still consistent with $\mu$P. In Figure S.16 , we trained 3-layer MLP with damping determined by Eq. 7. Its learning rate is fixed at 0.01. $\rho'_A$ and $\rho'_B$ are tuned using the following grid:

- $\rho'_A = 2^z$, where $z \in \{0, 1, 2, ..., 10\}$

| | Epochs=100 | | Epochs=200 | | Epochs=300 | |
|---|---|---|---|---|---|---|
| | K-FAC | Shampoo | K-FAC | Shampoo | K-FAC | Shampoo |
| wid | SP / $\mu$P | SP / $\mu$P | SP / $\mu$P | SP / $\mu$P | SP / $\mu$P | SP / $\mu$P |
| 1 | 68.53/+1.87 | 65.36/-0.31 | 71.55/+1.88 | 68.49/+0.30 | 72.28/+1.93 | 69.20/-0.02 |
| 2 | 71.28/+3.66 | 69.28/+0.54 | 73.85/+2.99 | 72.24/+0.11 | 74.30/+3.23 | 73.05/+0.20 |
| 4 | 69.45/+7.87 | 72.81/+0.78 | 73.39/+5.83 | 75.59/+0.38 | 73.79/+5.91 | 76.57/+0.31 |
| 8 | 61.15/+17.01 | 74.87/+0.28 | 68.13/+11.63 | 77.10/+1.64 | 68.56/+12.09 | 78.08/+1.74 |

**Figure S.14 : Test Accuracies of ResNet50 on CIFAR100.** We trained ResNet50 on CIFAR100 with $\eta = 0.003, \rho' = 10$. $\mu$P has a higher accuracy than SP in models with wide widths. Note that in the training of K-FAC, the standard deviation of the last layer at width=1 is tuned carefully and it is 1/16 of the default SP value.

| | VGG19 (C100) | | ResNet18 (C100) | | | |
|---|---|---|---|---|---|---|
| | Shampoo | | K-FAC | | Shampoo | |
| width | SP | $\mu$P | SP | $\mu$P | SP | $\mu$P |
| 1 | $63.30_{(0.26)}$ | $63.01_{(0.44)}$ | $67.02_{(0.33)}$ | $67.00_{(0.30)}$ | $67.01_{(0.33)}$ | $66.99_{(0.35)}$ |
| 2 | $70.10_{(0.23)}$ | $70.21_{(0.29)}$ | $71.88_{(0.36)}$ | $72.31_{(0.39)}$ | $70.83_{(0.18)}$ | $\mathbf{71.68_{(0.21)}}$ |
| 4 | $74.65_{(0.24)}$ | $75.04_{(0.18)}$ | $74.09_{(0.33)}$ | $\mathbf{75.06_{(0.22)}}$ | $73.80_{(0.26)}$ | $75.28_{(0.34)}$ |
| 8 | $76.51_{(0.38)}$ | $\mathbf{77.30_{(0.21)}}$ | $74.90_{(0.20)}$ | $\mathbf{76.92_{(0.12)}}$ | $76.23_{(0.21)}$ | $78.39_{(0.17)}$ |
| 16 | $77.89_{(0.18)}$ | $\mathbf{78.35_{(0.19)}}$ | $74.29_{(0.41)}$ | $\mathbf{78.12_{(0.25)}}$ | $78.05_{(0.21)}$ | $80.24_{(0.23)}$ |

**Table S.2 : Test accuracies with different widths (with error bar).** The results are averaged over 5 random seeds, with standard deviation shown in the brackets. The settings are the same as in Table.3.

- $\rho'_B = 2^z$, where $z \in \{0, -1, -2, ..., -10\}$

Figure S.16 shows that the optimal damping value for wide width models is given at $(d_A, d_B) = $ (-1, 1) in Eq. 7. This is consistent with the scaling in Eq. 9.

There are two reasons why we determine damping using Eq. 15 instead of Eq. 7. The first reason is that the optimal values for $\rho'_A$ and $\rho'_B$ are generally different, resulting in one extra hyperparameter. The second reason is that the eigenvalues of the curvature matrix may change over training and the optimal values for $\rho'_A$ and $\rho'_B$ may change during training(Zhang et al., 2019b). Figure S.16 shows that the mean eigenvalue of $A_l$ and $B_l$ change during the training and the damping determined by Eq. 15 and Eq. 14 is adjusted according to the change of mean eigenvalue.

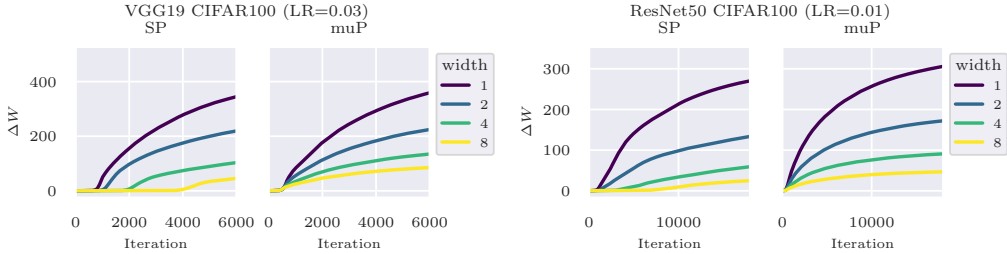

**Figure S.15 : For SP, $\Delta W/|W|$ in the early stages of training depends on the model width** In training with SP, $\Delta W/|W|$ depends on the width, especially in the early stages of training. Specifically, as the width increases, the period during which $\Delta W_l/|W_l| \approx 0$ gets longer. This suggests that in the infinite width limit with SP, $\Delta W/|W|$ remains 0 throughout training. On the other hand, in $\mu$P, the point at which weights start to move away from their initial values remains the same, even as the width increases (Left) We trained VGG19 on CIFAR100 using K-FAC. (Right) We trained ResNet50 on CIFAR100 using K-FAC.

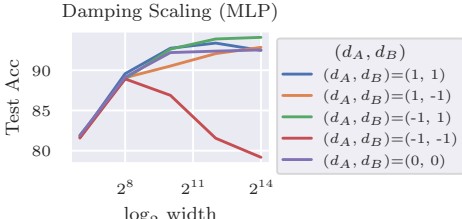

**Figure S.16 : Damping term needs to be scaled as** $d_A = -1$ **and** $d_B = 1$ **(Eq. 7).** We trained a 3-layer MLP on MNIST with K-FAC. The number of samples is reduced to 1024 and trained by MSE loss. Validation accuracy is highest at $d_A = -1$ and $d_B = 1$

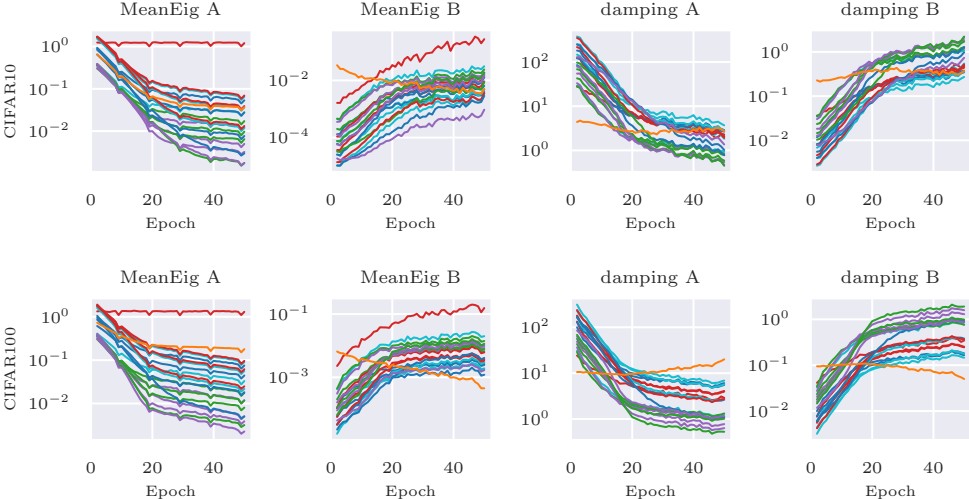

**Figure S.17 : Mean eigenvalues and the heuristics damping terms vary during training.** We trained ResNet18 on CIFAR10 and CIFAR100 by K-FAC. (Left) Transition of the mean eigenvalues of $A_l$ and $B_l$ in each layer during the training process. We can observe that the mean eigenvalues change significantly during the training. This implies that $\rho_A$ and $\rho_B$ need to be modified during the training to retain their second-order properties for numerical stability. Note that the transition of damping determined by Eq. 15 is consistent with the transition of mean eigenvalues. (Right) Transition of the damping determined by Eq. 14. The trend of damping determined by Eq. 14 is generally consistent with that of mean eigenvalues.

### H.2 DAMPING TRANSFER ON MLP

In Section 5.3, we have observed that when using heuristics damping, the optimal damping value shifts as the width increases. We have also confirmed that rescaling damping prevents this shift. This phenomenon is observed not only in CNN but also in MLP as shown in Figure S.18 . In the second column of Figure S.18 , the dataset size is not reduced, but again the damping is transferred by using rescaling damping.

### I LAZY PARAMETERIZATION

In the lazy regime, the network output changes by an order of $\Theta(1)$, but feature learning does not occur, i.e., $r_{l<L} > 0$. Consider the uniform parameterization (UP), which is easy to characterize and known to include two important parameterization methods; $\mu$P and NTK parameterization (Yang & Hu, 2021). It is defined by $r_{l<L} = r$ and $W_L$ satisfying Conditions A.1 and A.2. For K-FAC, we obtain $2a_L + c_L + e_A - 1 = 0, a_L + b_L = 1 - r$ from these conditions and thus

$$\begin{cases} 2a_1 + c_1 - e_B(2r - 1) = 2r - 1, \\ 2a_l + c_l + e_A - e_B(2r - 1) = 2r, \\ 2a_L + c_L + e_A - 1 = 0, \\ a_L + b_L = 1 - r. \end{cases} \tag{S.82}$$

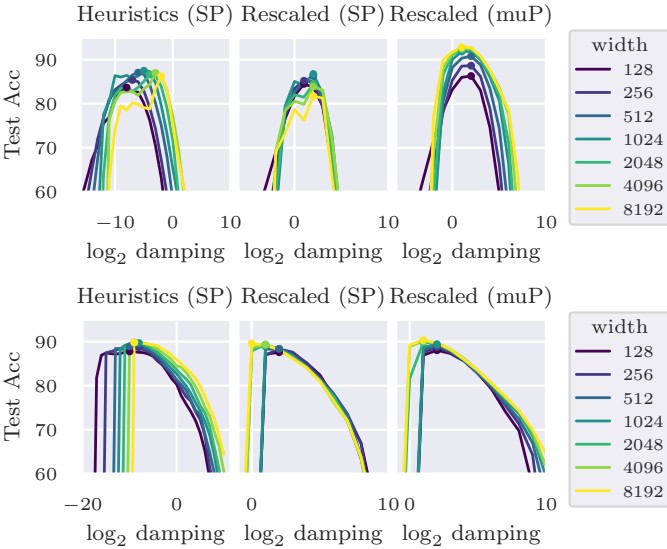

**Figure S.18 : We can transfer damping in MLP** We trained a 3-layer MLP on MNIST and a 3-layer MLP on FashionMNIST with K-FAC. When training MNIST, the number of samples is reduced to 1024 while in FashionMNIST we trained MLP by full-dataset. When using heuristics damping, the maximum damping value that does not diverge increases as the width increases.

We usually suppose $r = 1/2$ for NTK parameterization. By fixing the shift invariance by $a_l = 0$, we have

$$\begin{cases} c_1 = 0, & c_{l>1} = 1 - e_A, \\ b_1 = 0, & b_{l>1} = 1/2. \end{cases} \tag{S.83}$$

This means that K-FAC in SP achieves the lazy regime for the constant learning rates of $\Theta(1)$. In contrast, the first-order gradient ($e_A = 0$) in SP requires the re-scaled learning rates of $\Theta(1/M)$ for the lazy regime. Given that the naive setting of default hyperparameters in implementation often assumes SP and constant learning rates, it can be said that K-FAC is more likely to suffer from the lazy regime compared to (S)GD. It is also noteworthy that the obtained abc-parameterization is consistent with the parameterization used in parameterization used in the previous work on K-FAC in the NTK regime (Karakida & Osawa, 2020).

Similarly, for Shampoo, we obtain

$$\begin{cases} c_1 = 0, & c_{l>1} = 1 - e, \\ b_1 = 0, & b_{l>1} = 1/2. \end{cases} \tag{S.84}$$

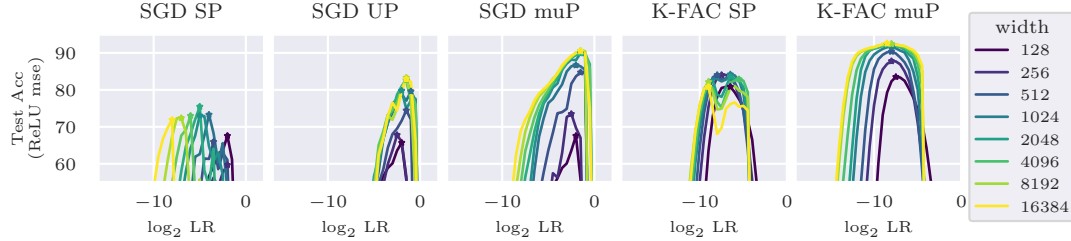

**Figure S.19 : UP allows the learning rate to transfer while the accuracy of UP is lower than $\mu$P.** We trained a 3-layer MLP on MNIST and the sample size was reduced to 1024. Note that in K-FAC SP is equal to the UP.

**Table S.3 : Lazy parameterization for SGD, K-FAC, FOOF and Shampoo**

|         | Input weights & all biases | Output weights | Hidden weights |
|---------|---------------------------|----------------|----------------|
| SGD     | $b=0, c=0$ | $b=1/2, c=1$ | $b=1/2, c=1$ |
| Shampoo | $b=0, c=0$ | $b=1/2, c=1/2$ | $b=1/2, c=1/2$ |
| K-FAC   | $b=0, c=0$ | $b=1/2, c=0$ | $b=1/2, c=0$ |
| FOOF    | $b=0, c=0$ | $b=1/2, c=0$ | $b=1/2, c=0$ |

