# OpenReview forum: "On the Parameterization of Second-Order Optimization Effective towards the Infinite Width"
_ICLR.cc/2024/Conference — ICLR 2024 poster_

### Official Review · Reviewer_SW4H · 2023-10-31

**Soundness:** 3 good
**Presentation:** 3 good
**Contribution:** 3 good
**Rating:** 6
**Confidence:** 3

**Summary:**

This paper considers finding the optimal hyperparameters for two second-order methods KFAC and Shampoo when training a neural network with a large width or when the width is infinite. These hyperparameters include the learning rate and the initialization hyperparameters. This optimal setting could help the network to learn features in a stable manner. This work is built upon the previous works that find the optimal setting for the first-order optimization methods.

**Strengths:**

Analyzing the optimal hyper-parameters for the second-order methods is provided in this work for the first time. Empirically they show that their suggested values for the hyperparameters help to find useful features when they increase the width of the layers i.e. their experimental results confirm their theoretical analysis. As a side contribution, they show that for a specific parameter setting KFAC has an implicit bias toward NNGP.

**Weaknesses:**

The paper mentions some conditions for the stable feature learning that comes from the previous works. However, I think it needs more context for a reader who is not aware of previous work.
In the experiments, for example in Table 3, there isn’t any standard deviation next to the accuracies.

**Questions:**

I was wondering if we change the layer width for different layers what part of your analysis would change?

---

> ### Author Response · Authors · 2023-11-21
> **Response to Reviewer SW4H**
>
> We would like to thank the reviewer for the helpful comments and suggestions.
>
> > In the experiments, for example in Table 3, there isn’t any standard deviation next to the accuracies.
>
> Thank you for the valuable advice.
> We have added data with standard deviations for Table 3 in Appendix G.2 (Table S.2).
> We initially avoided showing the standard deviation to prevent the tables and figures from becoming overly dense and impairing readability.
> However, since the purpose of Table 3 is the comparison of accuracy, it will be indeed beneficial to include standard deviations. Thus, we have now incorporated them in Table S.2 and it clarifies that the superiority of $\mu$P increases for wider networks.
> Please note that not all data have standard deviations due to limited time for rebuttal.
>
> > The paper mentions some conditions for the stable feature learning that comes from the previous works. However, I think it needs more context for a reader who is not aware of previous work.
>
> Thank you for your kind suggestion.
> We have added an additional explanation of the conditions to Appendix A.1.
> We hope you will also refer to Appendix A.2, which describes conditions A.1 and A.2 in more detail.
>
> > I was wondering if we change the layer width for different layers what part of your analysis would change?
>
> Thank you for your insightful question. Actually, if all widths approach infinity in the same order, our results remain the same!
> In more detail, if we define the model width as $M_l = \alpha_l M$ and consider the limiting case where $M$ becomes infinitely large while the coefficients $\alpha_l$(>0) remain constant, the obtained $\mu$P is the same.
> This is because the $\mu$P is based on order evaluation and the constant coefficients play no essential role.
> We have added an explanation of this case in the footnote of Appendix A.1.
>
> If our response satisfies the reviewer, we appreciate it if you would consider adjusting their score accordingly.

---

### Official Review · Reviewer_Xrk9 · 2023-11-02

**Soundness:** 3 good
**Presentation:** 3 good
**Contribution:** 3 good
**Rating:** 8
**Confidence:** 5

**Summary:**

This paper is a continuation of [1] in the context of second-order optimization. The initial idea consists in tuning simultaneously several hyperparameters (HPs) related to neural network (NN) training based on the following heuristic: after one training step, the pre-activations of each layer should have made a move of order $1$. More specifically, this heuristic should hold when the number of neurons of each layer tends to infinity. In order-$1$ optimization, the HPs include: variance of the initialization distribution of the weights, scaling of the weights, scaling of the learning rate. In the present work, the authors propose to tune the HPs specific to order-$2$ optimization, such as the damping parameters.

In practice, the authors focus on two order-$2$ optimization techniques (K-FAC and Shampoo), and test experimentally their heuristic with multilayer perceptrons (MLPs), convolutional NNs (CNNs) and ResNets. The authors find out that their method leads to better test losses, especially for NNs with very large layers. Moreover, one choice of learning rate seems to lead to results stable w.r.t. the size of the layers.

[1] *Tensor programs IV: Feature learning in infinite-width neural networks*, Yang and Hu, 2021.

=== Edit ===
I acknowledge the other reviews and the authors' answers. I keep my rating (8: Accept).

**Strengths:**

## Originality

Generalizing the heuristic of [1] to second order methods is new.

## Clarity

Overall, the paper is clear for a reader who is already aware of the works of Greg Yang (see [1]).

## Quality

* This work is technically correct.
* The initial heuristic is tested (Fig. 1): $\Delta h$ behaves as expected when using $\mu \mathrm{P}$.
* In order to compare fairly $\mu \mathrm{P}$ with SP, the authors have performed a sanity check (Fig. 5): the results are stable for a wide range of learning rates.
* The experiments show that $\mu \mathrm{P}$ is better than SP, especially when the number of neurons per layer tends to infinity. This result was expected and is explained by the "infinite width limit assumption".

**Weaknesses:**

## Section 5.2 and Figure 6

There is a mix of lack of clarity and unconvincing results in this section.

First, the authors claim in Section 5.2 that $\mu \mathrm{P}$ allows better "transfer" of learning rate than SP. The term "transfer" is not so clear to me, but apparently, it means that the optimal learning is stable by width change. But, when looking at Figure 6, this is barely true: actually, the transfer is better with K-FAC SP than with $\mu \mathrm{P}$ in the MLP case.

Second, in the legend of Figure 6, the authors claim something else: according to the context, "transferring the learning rate" means that, with a fixed l.r., the results get better and better as the width of the layers increases. This claim is more convincing, but differs substantially from the claim provided in Section 5.2.

## Significance

The narrowness of the focus on K-FAC and Shampoo is a limitation of this work. What do the authors think about Newton, Gauss-Newton and generalized Gauss-Newton?

Besides, the results presented in Table 1 are somewhat disappointing: basically, the usual NTK scaling corresponds to $\mu \mathrm{P}$ for all the hidden weights, in all the settings (SGD, Shampoo, K-FAC). The only difference lies in the input and output weights. Naturally, the authors cannot do anything about that.

**Questions:**

What do the authors think about apply $\mu \mathrm{P}$ to other second order methods, such as Newton, Gauss-Newton and generalized Gauss-Newton?

Could the authors clarify the meaning of "transferring the learning rate" in Section 5.2 and in Figure 6, and clarify the claims of the corresponding paragraphs?

---

> ### Author Response · Authors · 2023-11-21
> **Response to Reviewer Xrk9**
>
> We would like to thank the reviewer for the helpful suggestions and for correctly acknowledging the strengths of our work.
>
> > What do the authors think about apply to other second order methods, such as Newton, Gauss-Newton and generalized Gauss-Newton?
>
> Thank you for the insightful question.
> In deep learning, for standard training objectives such as MSE loss and cross-entropy loss, the Gauss-Newton matrix is equivalent to the Fisher Information Matrix [1].
> Thus, K-FAC, which is an approximation method for Natural Gradient Descent, can also be regarded as an approximation method for the Gauss-Newton method.
>
> Furthermore, we can derive the $\mu$P for the Gauss-Newton method in a similar way to K-FAC and Shampoo.
> Through some calculations, we found that the $\mu$P for the (layerwise) Gauss-Newton method is the same as in Proposition 4.1. In more detail, $b_l$ and $c_l$ are the same as in K-FAC and damping terms are the same as in Shampoo. Thus, the current work gives a unified perspective covering various individually developed second-order methods.
> We have included an explanation for this in Appendix A.4.3.
>
> [1] New insights and perspectives on the natural gradient method, James Martens, 2020
>
> > Could the authors clarify the meaning of "transferring the learning rate" in Section 5.2 and in Figure 6, and clarify the claims of the corresponding paragraphs?
>
> Thank you for your very helpful remarks. We should emphasize that in the previous work of $\mu$P, learning rate transfer is defined as satisfying both "we can set the optimal learning rate without depending on the order of width" and "wider is better".
> Since our explanation was insufficient at this point, we have added this definition in Section 5.2.
>
> > But, when looking at Figure 6, this is barely true: actually, the transfer is better with K-FAC SP than with muP in the MLP case.
>
> Thank you for the question.
> In Figure 6, the learning rate does not transfer in SP.
> More specifically, when training MLP with K-FAC SP, the optimal learning rate for width = 16384 is lower than that for width = 128 (which is marked by star symbols in the figure).
> Consequently, if we use the optimal learning rate determined for a width of 128 to train a model with a width of 16384, there is a significant drop in accuracy compared to using the optimal learning rate for the latter width. In contrast, in the K-FAC $\mu$P, we can transfer a learning rate, allowing the model with width = 16384 to be effectively trained using the optimal learning rate for width = 128. Furthermore, "wider is better" holds as well.

---

> > ### Comment · Reviewer_Xrk9 · 2023-11-22
> >
> > For the question of "learning rate transfer", I formally agree with the remark of the authors. However, the range of working learning is moving as the width increases. And this translation is more noticeable in this configuration than with K-FAC SP (at least for the MLP and the CNN). More precisely, the interval of working LRs is expanding on the "small learning rates" side (by one order of magnitude).
> >
> > Even if this fact is subtle, it is not explained in the paper. Anyway, I do not request any change about it (but it should be kept in mind for future research).

---

### Official Review · Reviewer_ZUFF · 2023-11-07

**Soundness:** 3 good
**Presentation:** 3 good
**Contribution:** 3 good
**Rating:** 8
**Confidence:** 3

**Summary:**

The authors derive scaling laws for two second-order optimization methods (K-FAC (Martens & Grosse, 2015) and Shampoo (Gupta et al, 2018)), based on the muP approach from (Yang & Hu, 2021). This approach has previously been used for first-order methods, including with momentum. It is used to determine how the weight initialization and per-layer learning rate for deep neural networks should scale based on layer width. This allows 'parameter transfer', where good hyperparameters can be trained on small networks and suitably scaled up for wider networks. The derived scalings are successfully tested on some simple example problems.

**Strengths:**

This is a new application of the muP framework to a useful class of methods. I think the new scaling laws are useful and beneficial to the community. The authors apply a successful approach for deriving scaling laws to the case of specific second-order networks. The results are clearly stated and yield new insights into the merit of existing choices of damping parameter for K-FAC, and the use of zero initial weights for the last layer. The paper is reasonably clearly written and the presented experiments thorough.

**Weaknesses:**

I found the mathematical presentation fairly informal, both in terms of the exposition of the approach (Prop 4.1 has "the optimization becomes valid for...", for example) and the actual derivation in Appendix A. This is a presentational comment, not a critique of the correctness of the work.

Theory is only given for least-squares losses, although the numerics suggest the results hold for other losses. If the authors could derive these laws - and Appendix A seems to suggest it might not be a huge piece of work - this would help the paper.

While the empirical results show merit, their real benefit would only be clear if applied for larger-scale networks than those tested. I understand the practical issues around doing this and recognize that it isn't feasible for all researchers, but those results would strengthen the paper.

=========== AFTER REBUTTAL ===========
The new sections on layer-wise Gauss-Newton and extension to other loss functions are helpful here. I would suggest the authors mention the Gauss-Newton work somewhere in the main text (just like they did for the alternative loss functions), as I couldn't see clear reference to this extra contribution.

**Questions:**

To me, it appears that the extent of the theoretical contribution is not huge (largely based on combining existing methods), and I take the main benefit of the paper to be the actual derived scaling laws (i.e. choices of $a$, $b$, $c$, etc) rather than the overall approach/techniques.  Does this work introduce any new theoretical results/mathematical techniques of wider interest/applicability, or is the theoretical approach quite restricted to the specific methods being considered?

=========== AFTER REBUTTAL ===========
The new version of the work includes a mild strengthening of the mathematical formatlity in Appendix A, plus new results deriving scaling laws for layer-wise Gauss-Newton and other loss functions such as cross-entropy. This I think improves the paper by showing that their adaptation of muP to second-order methods yields a systematic approach rather than needing ad-hoc approaches, giving it wider interest.

---

> ### Author Response · Authors · 2023-11-21
> **Response to Reviewer ZUFF**
>
> We would like to thank the reviewer for the helpful comments and suggestions.
>
> **Answer to Question**
> >  I take the main benefit of the paper to be the actual derived scaling laws ... Does this work introduce any new theoretical results/mathematical techniques of wider interest/applicability...?
>
> As you appropriately understand, the main benefit of this paper is the actual derived scaling laws.
> However, we think that our theoretical approach also includes some novelty in the theory. First,
> we provided a *unified* derivation that is easier to apply to 1-st order graident compared to the previous work and also applicable to our 2nd order gradients. In more detail,
> as is summarized in the beginning of Section A.2, we showed the proof focusing on the order evaluation of $h_{l,1}$ rather than the kernel value $||h_{l,1}||^2$. Indeed, this does not involve developing new mathematical tools, so from that perspective, the novelty might be low. However, we believe that formulating derivations in a unified manner that is easily applicable to various problems is also a significant contribution in theory.
>
> Second, a technical novelty in applying muP to second-order optimization is the usage of push-through identity(Lemma A.7). This gives a unified derivation over K-FAC and Shampoo. Note that it also covers (layer-wise) Gauss-Newton gradient, which was requested by Reviewer Xrk9 and has been added in Section A.4.3.
> As far as we know, these 2nd-order methods have been analyzed only in an individual manner.
> Therefore, it is interesting that we can show new theoretical results and techniques that are widely applicable to these seemingly different methods.
>
> **Reply to Weakness**
> >  Theory is only given for least-squares losses, although the numerics suggest the results hold for other losses.
>
> Thank you for your valuable suggestion.
> As you say, the results of Proposition 4.1 in this paper can easily be extended to the cross-entropy loss.
> Technically speaking,  we need to change only the definition of $\chi$ in (S.21) and the coefficient matrix of order 1 appearing in $B_l$ matrices, which does not affect the order of $\delta_l$ and $h_l$.
> We have added an explanation in Appendix A.5 that the muP is the same among all MSE and cross-entropy losses.
>
> > I found the mathematical presentation fairly informal... This is a presentational comment, not a critique of the correctness of the work.
>
> Our focus was the derivation of the actual scaling, so we prioritized a structure that is somewhat informal but more accessible to readers outside of mathematics, rather than adhering strictly to a mathematically formal composition (writing everything in the format of definitions, lemmas, propositions, and theorems).
> However, we agree that at least we need to clearly mention any assumption or new definition unique to our work. To avoid potential misunderstanding in the main text, we have added references to the assumptions used in the Appendix. We have also added a detailed explanation of the 'valid', as defined by us, in the derivations within the Appendix (Definition A.6).
>
> > While the empirical results show merit, their real benefit would only be clear if applied for larger-scale networks than those tested.
>
> We also agree that doing more experiments on large-scale networks will further verify the benefit of this work. We hope that the appearance of the current work will inspire the community and subsequent work will challenge individual large-scale networks including Transformers as is mentioned in Section 6.
>
> If our response satisfies the reviewer, we appreciate it if you would consider adjusting their score accordingly.

---

> > ### Author Response · Authors · 2023-11-23
> > **Response to reviewer feedback**
> >
> > We deeply appreciate raising the score and providing valuable advice. Based on your suggestion, we add a mention of $mu$P for the Gauss-Newton method in Section 4.1.

---

### Author Response · Authors · 2023-11-21
**We thank all the reviewers again for their valuable feedbacks.**

We are pleased to see that all reviewers, after their kind reading, have provided accept-side evaluations and have also offered helpful suggestions to further improve our manuscript. We have revised our paper accordingly to the reviewers' suggestions, and they are marked in red for clarity.

---

### Meta-Review · Area_Chair_KWn7 · 2023-12-02

**Metareview:**

The paper derives appropriate scales of hyperparameters such as random initialization, learning rates, and damping terms for two major second-order optimization algorithms, namely K-FAC and Shampoo. The paper then experimentally demonstrates the affects of the proposed results on several learning tasks.

The overall sentiment among the reviewers is that the paper makes novel contributions to extend the muP framework to some members of the class of second-order optimization methods, and the results are significant and useful for the community.

**Justification For Why Not Higher Score:**

The results are novel but the scope is limited to only two optimization methods and as a result is ad-hoc and not general. The experimental section could be greatly improved.

**Justification For Why Not Lower Score:**

The results are novel and useful enough to warrant acceptance.

---

### Decision · Program_Chairs · 2024-01-16

Accept (poster)